# Modulation of NMDA receptor signaling and zinc chelation prevent seizure-like events in a zebrafish model of SLC13A5 epilepsy

Deepika Dogra[1,2,3], Van Anh Phan[1,2,3], Sinan Zhang[1,2,3], Cezar Gavrilovici[1,2,3,4], Nadia DiMarzo[1,2,3], Ankita Narang[2], Kingsley Ibhazehiebo[1,2,3], Deborah M. Kurrasch[1,2,3]*

1 Department of Medical Genetics, University of Calgary, Calgary, Alberta, Canada, 2 Alberta Children's Hospital Research Institute, University of Calgary, Calgary, Alberta, Canada, 3 Hotchkiss Brain Institute, University of Calgary, Calgary, Alberta, Canada, 4 Departments of Pediatrics, Clinical Neurosciences, Physiology & Pharmacology, Cumming School of Medicine, University of Calgary, Calgary, Alberta, Canada

* kurrasch@ucalgary.ca

## Abstract

*SLC13A5* encodes a citrate transporter highly expressed in the brain and is important for regulating intra- and extracellular citrate levels. Mutations in this gene cause rare infantile epilepsy characterized by lifelong seizures, developmental delays, behavioral deficits, poor motor progression, and language impairments. SLC13A5 individuals respond poorly to treatment options; yet drug discovery programs are limited due to a paucity of animal models that phenocopy human symptoms. Here, we used CRISPR/Cas9 to create loss-of-function mutations in *slc13a5a* and *slc13a5b*, the zebrafish paralogs to human *SLC13A5*. *slc13a5* mutant larvae showed cognitive dysfunction and sleep disturbances, consistent with SLC13A5 individuals. These mutants also exhibited fewer neurons and a concomitant increase in apoptosis across the optic tectum, a region important for sensory processing. Further, *slc13a5* mutants displayed hallmark features of epilepsy, including an imbalance in glutamatergic and GABAergic excitatory-inhibitory gene expression, increased *fosab* expression, disrupted neurometabolism, and neuronal hyperexcitation as measured in vivo by extracellular field recordings and live calcium imaging. Mechanistically, we tested the involvement of NMDA signaling and zinc chelation in *slc13a5* mutant epilepsy-like phenotypes. Slc13a5 protein co-localizes with excitatory NMDA receptors in wild-type zebrafish and NMDA receptor expression is upregulated in the brain of *slc13a5* mutant larvae. Additionally, low levels of zinc are found in the plasma membrane of *slc13a5* mutants. NMDA receptor suppression and ZnCl$_2$ treatment in *slc13a5* mutant larvae rescued neurometabolic and hyperexcitable calcium events, as well as behavioral defects. These data provide empirical evidence in support of the hypothesis that excess extracellular citrate over-chelates the zinc ions needed to regulate NMDA receptor function, leading to sustained channel opening and an exaggerated excitatory response that manifests as seizures. These data show the utility of *slc13a5*

**Data availability statement:** All relevant data are within the paper and its Supporting information files. Bulk RNA Sequencing data reported in this work are available through NCBI GEO (GSE275235). All codes used for analysis were uploaded to Zenodo and are available at https://zenodo.org/records/14853182, as mentioned in the Materials and Methods section.

**Funding:** This work was supported by Brain Canada Platform Support Grant (Grant Number- PSG DMK-2014 to DMK; https://braincanada.ca/), TESS Research Foundation Early-Career Investigator Research Grant (Grant Number- RMS22-81166867 to DD; https://www.tessresearch.org/) and Owerko Centre Early Career Investigator Grant (Grant Number- 10022909 to DD; https://research-4kids.ucalgary.ca/owerko/owerko). The funders had no role in study design, data collection and analysis, decision to publish, or preparation of the manuscript.

**Competing interests:** The authors have declared that no competing interests exist.

**Abbreviations:** CC3, cleaved-Caspase 3; CTCF, corrected total cell fluorescence; DEE25, developmental epileptic encephalopathy 25; dpf, days post fertilization; FACS, Fluorescence-activated cell sorting; FDR, false discovery rate; HRMA, high-resolution melt analysis; IEG, immediate early genes; LMA, low melting agarose; MB, midbrain; NaCT, sodium-coupled citrate transporter; OCR, Oxygen consumption rate; PCA, principal component analysis; qPCR, quantitative PCR; WTs, wild-type siblings.

mutant zebrafish for studying SLC13A5 epilepsy and open new avenues for drug discovery.

## Introduction

SLC13A5 is a sodium-coupled citrate transporter (NaCT), encoded by the *SLC13A5* gene in mammals and expressed in the plasma membrane of liver, testis, bone, and neural cells [1–4]. In the human brain, the citrate transporter is expressed predominantly in neurons, where it mediates the uptake of circulating and astrocyte-released citrate. Citrate plays a crucial role in lipid and cholesterol biosynthesis, as well as in the regulation of energy expenditure [5]. Mutations in *SLC13A5* cause developmental epileptic encephalopathy 25 (DEE25) characterized by prolonged and frequent seizures as early as the first day of life [6–10]. Other associated comorbidities include developmental delays, teeth hypoplasia, cognitive impairments, sleeping difficulties, and slow motor progress [11–13]. Its *Drosophila* homolog, *Indy*, initially was associated with longevity [14] but upon re-examination was found to exhibit a loss of glutamatergic neurons and an overlooked "bang-induced" seizure-like phenotype [15]. *Slc13a5* knockout mice are small compared to wild-type controls and display metabolic phenotypes such as protection against obesity and insulin resistance [16]. Although originally thought to lack any epilepsy-related phenotypes, more recently *Slc13a5*-null animals were found to display neural hyperexcitability but no obvious behavioral or histological abnormalities [17], questioning the utility of mice in modeling the human SLC13A5 syndrome given the strong seizures experienced. Further, Dirckx and colleagues now report an osteoblast metabolic pathway dysregulated in *Slc13a5*$^{-/-}$ mice, revealing insights into the teeth and bone deformations observed in humans [18].

Individuals suffering from SLC13A5 epilepsy harbor either homozygous or compound heterozygous *SLC13A5* pathogenic variants. Antiseizure medications such as stiripentol or a combination of acetazolamide and valproic acid reduce seizure frequency in some SLC13A5 individuals; however, most children are refractory and complete seizure freedom is never attained even in those that do respond to anti-seizure medications [13]. In vitro functional studies on several *SLC13A5* mutations have revealed disrupted transporter activity consistent with a loss of function [6,19], but the molecular mechanisms contributing to seizures are still poorly understood.

Zebrafish are a powerful system for studying epilepsy with several genetic and pharmacologically induced models recapitulating human epileptiform activity [20–24]. Moreover, zebrafish larvae are a useful vertebrate model for understanding the etiology of neurological disorders given the range of behavioral assessments, genetic manipulations, and high-resolution measures of brain activity. To explore epilepsy-related phenotypes due to the loss of *SLC13A5* and study the molecular mechanisms behind this disorder, we generated the genetic mutants of both zebrafish *SLC13A5* paralogs, *slc13a5a* (referred to as *5a*) and *slc13a5b* (referred to as *5b*). Here, we characterized the phenotypes of all three mutants: *5a*$^{-/-}$ singly, *5b*$^{-/-}$ singly, and *5a*$^{-/-}$;*5b*$^{-/-}$ double mutants. Additionally, we studied the role of NMDA

receptor signaling and zinc chelation in *SLC13A5* epileptic phenotypes. Of note: to simplify the language throughout this text, hereafter, we use "*slc13a5$^{-/-}$*" or "*slc13a5* mutants" to refer to any one of these three mutant strains. The exact mutant analyzed is defined in the figures.

## Results

### *slc13a5* is expressed in the brain and *slc13a5$^{-/-}$* larvae display a smaller midbrain and increased mortality

Due to an ancestral gene duplication event, zebrafish possess two paralogs of *SLC13A5*: *5a* and *5b*. Using antisense RNA probes, we performed in situ hybridization for *5a* and *5b* expression and found that both paralogs were highly expressed in the zebrafish brain. Widespread expression was observed across the central nervous system during the early developmental stages of 25 hours post fertilization (hpf) and 3 days post fertilization (dpf); however, this expression perhaps became slightly stronger in the midbrain (MB) by 5 dpf (Fig 1A–1F). We further confirmed the specificity of the above-mentioned probes by performing in situ hybridization with sense RNA (control) probes at 3 dpf and found negligible expression of both *5a* and *5b* (S1A and S1B Fig). To examine the neural cell types expressing Slc13a5, we performed co-immunostaining for Slc13a5 with HuC/D (pan neuronal marker), Gfap (astrocyte marker), vGlut1 (excitatory neuron marker) or Gad67 (inhibitory neuron marker). We observed that Slc13a5 is expressed both in the neurons (including excitatory and inhibitory) and astrocytes in the MB region of 3 dpf zebrafish larvae, with a significantly higher percentage of neurons (85.5%) expressing Slc13a5 compared to astrocytes (14.2%) (Fig 1G–1K).

Next, we injected CRISPRs targeting the fifth exons of *5a* and *5b* genes to generate mutant alleles, *5a$^{-/-}$* and *5b$^{-/-}$*. *5a$^{-/-}$* allele mutations comprised an 11-nucleotide deletion and *5b$^{-/-}$* allele mutations comprised one-nucleotide substitution followed by 13-nucleotide insertion, both predicted to encode truncated proteins due to a premature stop (Fig 1L–1O). These mutations were targeted in the sodium-binding site of NaCT, conserved among humans and zebrafish (Fig 1N and 1O). Notably, several variants of SLC13A5 epilepsy have mutations in the sodium-binding site of NaCT, suggesting that sodium binding deficiency causes impaired citrate transport [7]. To assess gene expression levels and potential gene compensation, we used quantitative PCR (qPCR) at 5 dpf and showed that *5a* transcript levels were significantly reduced in *5a$^{-/-}$* and *5a$^{-/-}$;5b$^{-/-}$* larvae compared to wild-type siblings (WTs), indicating active mRNA degradation of *5a$^{-/-}$* transcripts. Notably, *5a* transcript levels were unchanged in *5b$^{-/-}$* single mutant larvae showing an absence of genetic compensation by paralog upregulation (S1C Fig). Similarly, qPCR analysis showed that *5b* transcript levels were significantly reduced in *5b$^{-/-}$* and *5a$^{-/-}$;5b$^{-/-}$* larvae compared to WTs at 5 dpf, also showing active mRNA degradation of *5b$^{-/-}$* transcripts, and unchanged in *5a$^{-/-}$* single mutant larvae, likewise indicating an absence of genetic compensation by paralog upregulation (S1D Fig). We next obtained neuronal population by performing fluorescence-activated cell sorting (FACS) on the heads of 5 dpf WTs *(Tg(elavl3:ubci-Cer-sv40))*. The significantly enriched expression of *elavl3* (also known as *huc*) in the sorted cells compared to astrocytic *gfap* confirmed the purity of the neuronal population (S1E Fig). Our qPCR analyses on these sorted neurons showed that both *5a* and *5b* transcript levels are significantly reduced in *5a$^{-/-}$;5b$^{-/-}$* larvae, further indicating that both the paralogs are highly expressed in the neurons compared to astrocytes (S1F and S1G Fig). Additionally, a significant reduction in the Slc13a5 fluorescence intensity, as measured by calculating corrected total cell fluorescence (CTCF) in the brains of Slc13a5 immunostained *5a$^{-/-}$;5b$^{-/-}$* larvae compared to WTs at 5 dpf, was observed, indicating the loss of Slc13a5 expression also at protein level (S1H–S1J Fig).

No significant gross morphological differences were observed between *slc13a5* mutants and WTs at 5 dpf (Fig 1P–1S). To examine neural changes, we performed immunostaining for the neuronal marker HuC/D and quantified the size of all three brain regions at 5 dpf by measuring the width of brain regions. *slc13a5$^{-/-}$* larvae exhibited −5.5% (*5a$^{-/-}$*), −4.9% (*5b$^{-/-}$*) and −6.3% (*5a$^{-/-}$;5b$^{-/-}$*) reduction in the MB size compared to WTs, with the forebrain (FB) and hindbrain (HB) size unchanged (Fig 1T-W). We further observed −29.8% (*5a$^{-/-}$*), −17.8% (*5b$^{-/-}$*) and −28.1% (*5a$^{-/-}$;5b$^{-/-}$*) reduction in MB

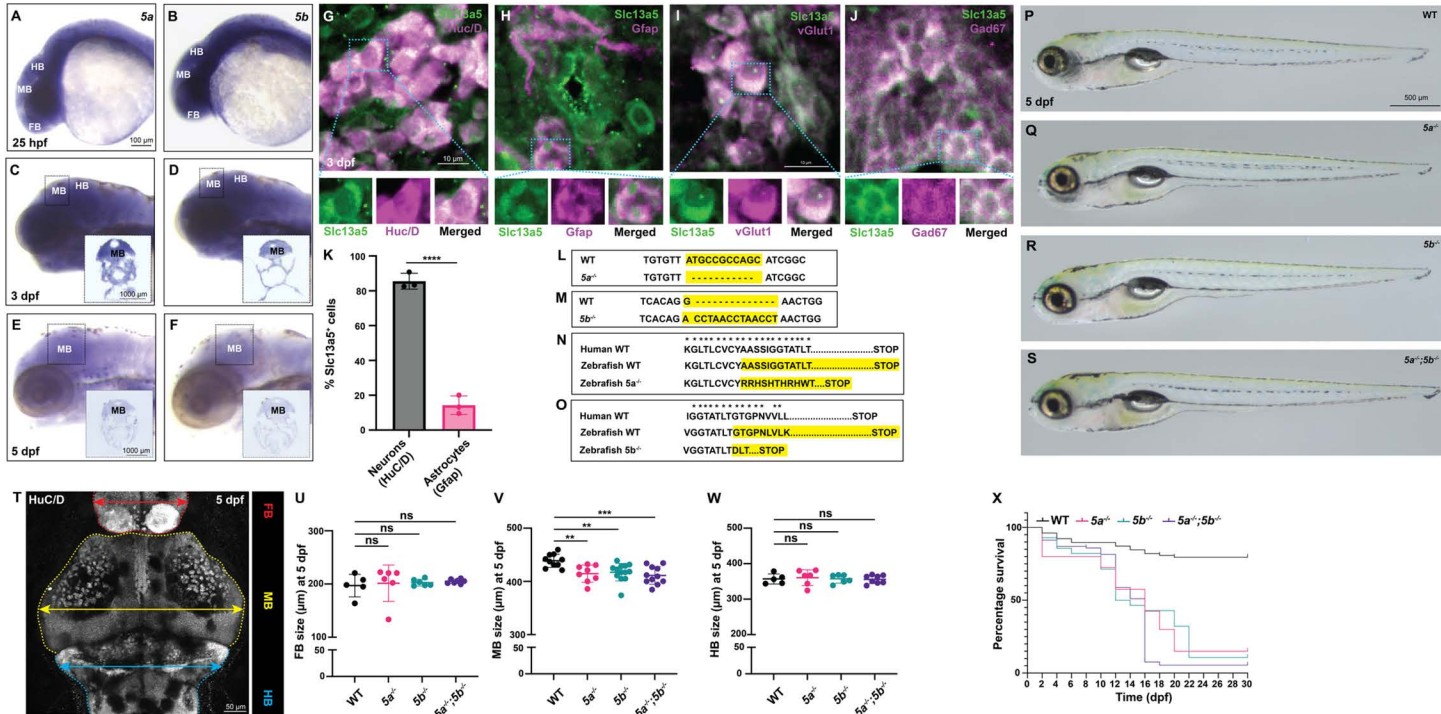

**Fig 1. Expression analysis of *slc13a5* zebrafish paralogs (*5a* and *5b*) and generation of zebrafish *slc13a5* mutants using CRISPR/Cas9 technique. (A**-F) In situ hybridization for *5a* and *5b* expression at 25 hpf, 3 dpf and 5 dpf using antisense RNA probes. Insets show sections of 3 dpf and 5 dpf larvae post in situ hybridization for *5a* and *5b* expression. *5a* and *5b* are expressed in all the brain regions during early development and their expression appears to get slightly stronger in the MB at later stages. (G) Section of 3 dpf WTs; α-Slc13a5, α-HuC/D. Insets of dashed box in G showing example of a cell expressing Slc13a5, HuC/D and merged signal. MB neurons express Slc13a5 (Slc13a5+/HuC/D+). (H) Section of 3 dpf WTs; α-Slc13a5, α-Gfap. Insets of dashed box in H showing an example of a cell expressing Slc13a5, Gfap and merged signal. A small population of MB astrocytes express Slc13a5 (Slc13a5+/Gfap+). (I) Section of 3 dpf WTs; α-Slc13a5, α-vGlut1. Insets of dashed box in I showing example of a cell expressing Slc13a5, vGlut1 and merged signal. MB excitatory neurons express Slc13a5 (Slc13a5+/vGlut1+). (J) Section of 3 dpf WTs; α-Slc13a5, α-Gad67. Insets of dashed box in J showing example of a cell expressing Slc13a5, Gad67 and merged signal. MB inhibitory neurons express Slc13a5 (Slc13a5+/Gad67+). (K) Percentage of Slc13a5+ neurons (HuC/D+) and astrocytes (Gfap+) in the MB of 3 dpf WT. There are significantly more Slc13a5+/HuC/D+ cells compared to Slc13a5+/Gfap+ cells. n = 3 for each group. (L) Nucleotide sequences of *5a* in zebrafish WT and *5a−/−*. Yellow highlighted regions indicate 11-nucleotide deletion due to mutation in *5a*. (M) Nucleotide sequences of *5b* in zebrafish WT and *5b−/−*. Yellow highlighted regions indicate one-nucleotide substitution followed by 13-nucleotide insertion due to mutation in *5b*. (N) Amino acid sequence of *5a* in human orthologue, zebrafish WT and *5a−/−*. Asterisks indicate conserved amino acids between human and zebrafish. Yellow highlighted regions indicate affected amino acids and premature stop due to mutation in *5a*. (O) Amino acid sequence of 5b in human orthologue, zebrafish WT and *5b−/−*. Asterisks indicate conserved amino acids between human and zebrafish. Yellow highlighted regions indicate affected amino acids and premature stop due to mutation in *5b*. **(P-**S) WT and *slc13a5* mutant larvae at 5 dpf. No morphological differences were observed. WT, n = 5; *5a−/−*, n = 5; *5b−/−*, n = 5; *5a−/−;5b−/−*, n = 5. **(T-**W) Quantification of FB, MB and HB size at 5 dpf by measuring brain width. MB size is significantly reduced in *slc13a5* mutants. FB and HB size is unchanged. WT, n = 5; *5a−/−*, n = 6; *5b−/−*, n = 6; *5a−/−;5b−/−*, n = 7 for FB size measurement. WT, n = 10; *5a−/−*, n = 8; *5b−/−*, n = 13; *5a−/−;5b−/−*, n = 11 for MB size measurement. WT, n = 5; *5a−/−*, n = 6; *5b−/−*, n = 6; *5a−/−;5b−/−*, n = 8 for HB size measurement. Colored lines with arrows on both ends show how the width measurements were performed. (X) Survival curve of WTs and *slc13a5* mutants. Very few *slc13a5* mutants survive until 30 dpf. WT, n = 78; *5a−/−*, n = 40; *5b−/−*, n = 28; *5a−/−;5b−/−*, n = 92. Data are Mean ± S.D., ns: no significant changes observed, **P ≤ 0.01, ***P ≤ 0.001, ****P ≤ 0.0001- Unpaired *t* test. hpf, hours post fertilization; dpf, days post fertilization; WT, wild-type siblings; *5a*, *slc13a5a*; *5b*, *slc13a5b*; FB, forebrain; MB, midbrain; HB, hindbrain. The data underlying this figure can be found in S1 Data.

volume of 5 dpf *slc13a5* mutants compared to WTs (S1K Fig). Consistently, the effect of *slc13a5* mutations on MB size correlates with the perhaps slightly stronger expression of both paralogs in the MB at a later zebrafish developmental stage. Next, we performed a survival curve analysis (0–30 dpf) and observed a drastic mortality of *slc13a5−/−* larvae, with only 15% (*5a−/−*), 10.7% (*5b−/−*) and 5.4% (*5a−/−;5b−/−*) mutants surviving past the juvenile stages and into adulthood at 30 dpf (Fig 1X).

**_slc13a5_<sup>−/−</sup> larvae exhibit behavioral deficits, reduced neuronal numbers, and increased neuronal apoptosis**

Individuals suffering from SLC13A5 epilepsy exhibit comorbidities, including cognitive impairments and sleep challenges [12,25]. To determine the effect of _slc13a5_ loss-of-function on zebrafish behaviors, we performed a 10-minute acoustic startle protocol by exposing 5 dpf larvae to 440 Hz vibration pulses in the light and calculated the total distance traveled (Fig 2A). The _slc13a5_<sup>−/−</sup> larvae moved longer distances compared to WTs, both during baseline and acoustic stimulation periods, with a significant increase in total distance moved during the complete protocol (413.2% _5a_<sup>−/−</sup>; 1213.3% _5b_<sup>−/−</sup>; 285.9% _5a_<sup>−/−</sup>;_5b_<sup>−/−</sup>), exhibiting hyperactive behavior of _slc13a5_ mutants in light without any external stimuli as well as their hypersensitivity to acoustic startle and a lack of short-term habituation, which can be a measure of learning potential and might resemble the cognitive dysfunction in SLC13A5 individuals (Figs 2B and S2A). The locomotion heatmaps further illustrate a hyperactivity phenotype of _slc13a5_<sup>−/−</sup> larvae compared to WTs when exposed to acoustic startle (Fig 2C). Next, we assessed the circadian behavior of _slc13a5_<sup>−/−</sup> larvae compared to WTs from 4 dpf to 6 dpf by tracking the total distance moved during 10 hours of darkness in the night and 6 hours of light in the daytime. The _slc13a5_<sup>−/−</sup> larvae traveled longer distances compared to WTs during night (4 dpf: 79.8% _5a_<sup>−/−</sup>; 192% _5b_<sup>−/−</sup>; 71.3% _5a_<sup>−/−</sup>;_5b_<sup>−/−</sup> and 5 dpf: 153.2% _5a_<sup>−/−</sup>; 231.3% _5b_<sup>−/−</sup>; 142.1% _5a_<sup>−/−</sup>;_5b_<sup>−/−</sup>), showing disruptions in nighttime behaviors (Fig 2D). The _slc13a5_<sup>−/−</sup> larvae also traveled longer distances compared to WTs during daytime (5 dpf: 177.3% _5a_<sup>−/−</sup>; 194.6% _5b_<sup>−/−</sup>; 157.2% _5a_<sup>−/−</sup>;_5b_<sup>−/−</sup> and 6 dpf: 209% _5a_<sup>−/−</sup>; 236.8% _5b_<sup>−/−</sup>; 211.4% _5a_<sup>−/−</sup>;_5b_<sup>−/−</sup>), further confirming a hyperactive phenotype during daytime (Fig 2D), which is consistent with other epileptic zebrafish models [26]. Approximately 20% of individuals with epilepsy suffer from anxiety [27,28], thus we tested the "wall-hugging" thigmotaxis behavior of our zebrafish mutants, a validated measure of anxiety in animals and humans [29–31]. We exposed the 5 dpf larvae to 6 min of light and 4 min of darkness and measured the distance they traveled and time they spent close to the wall, _i.e._, in the outer well zone (S2B Fig). There were no significant changes in the distance traveled and time spent close to the wall in mutants compared to WTs (S2C and S2D Fig). Some individuals with epilepsy are more sensitive to stimuli causing reflex seizures [32]. We thus challenged the 5 dpf larvae with an acute treatment of 5 mM pentylenetetrazol (PTZ) to induce instantaneous convulsions, followed by exposure to thigmotaxis protocol as described above. Under this condition, the _slc13a5_<sup>−/−</sup> larvae swam significantly longer distances (101.2% _5a_<sup>−/−</sup>; 75.8% _5b_<sup>−/−</sup>; 85% _5a_<sup>−/−</sup>;_5b_<sup>−/−</sup>) and spent significantly more time hugging the wall (110% _5a_<sup>−/−</sup>; 63.6% _5b_<sup>−/−</sup>; 70.3% _5a_<sup>−/−</sup>;_5b_<sup>−/−</sup>) compared to WTs, indicating anxiety-like behavior in the presence of a chemical stimulus (S2E and S2F Fig). Combined, _slc13a5_<sup>−/−</sup> larvae display a range of behavioral deficiencies.

We further studied whether these _slc13a5_<sup>−/−</sup> larval behavioral deficits correlated with any molecular neural changes. We focused our assessments on the optic tectum region of the MB as its size was significantly affected in our mutants. We performed immunostaining for HuC/D to quantify the number of neurons and observed a reduction in 5 dpf _slc13a5_<sup>−/−</sup> larvae compared to WTs (−27.5% _5a_<sup>−/−</sup>; −11.6% _5b_<sup>−/−</sup>; −22.3% _5a_<sup>−/−</sup>;_5b_<sup>−/−</sup>; Fig 2E and 2F). To determine when neuronal numbers started to decline, we quantified HuC/D positive-cells at 3 dpf and observed a decrease in _slc13a5_ mutants at this earlier stage (−37.7% _5a_<sup>−/−</sup>; −29.1% _5b_<sup>−/−</sup>; −19.9% _5a_<sup>−/−</sup>;_5b_<sup>−/−</sup>; Figs S3A and 2G), causing us to question if fewer neurons were born in _slc13a5_<sup>−/−</sup> to start. We quantified the number of neurons (HuC/D positive) around peak neurogenesis (28 hpf) in the region that gives rise to optic tectum and found no significant differences between _slc13a5_<sup>−/−</sup> larvae and WTs at this stage (S3B and S3C Fig). We further checked whether any alteration in cell proliferation at later stages is a possible cause of reduction in neuronal numbers and observed no significant changes in the number of pHH3<sup>+</sup> cells in 3 dpf and 5 dpf _slc13a5_<sup>−/−</sup> larvae compared to WTs (S3D-G Fig). We next questioned if cell death was a potential cause of the decreased neuronal numbers. We performed acridine orange (A.O.) staining at 5 dpf to label apoptotic cells and showed that _slc13a5_<sup>−/−</sup> larvae exhibited an increase in cell death (42.3% _5a_<sup>−/−</sup>; 33.5% _5b_<sup>−/−</sup>; 84.2% _5a_<sup>−/−</sup>;_5b_<sup>−/−</sup>) compared to WTs (Fig 2H and 2I). This result was further validated by performing co-immunostaining for HuC/D and cleaved-Caspase 3 (CC3), a marker for apoptotic cells, at 5 dpf where we observed an increase in the percentage of CC3<sup>+</sup> neurons in _slc13a5_<sup>−/−</sup> larvae compared to WTs (111.9% _5a_<sup>−/−</sup>; 120% _5b_<sup>−/−</sup>; 167.5% _5a_<sup>−/−</sup>;_5b_<sup>−/−</sup>), signifying that the reduction in neuronal numbers is a consequence of increased cell death (Figs S3H and 2J).

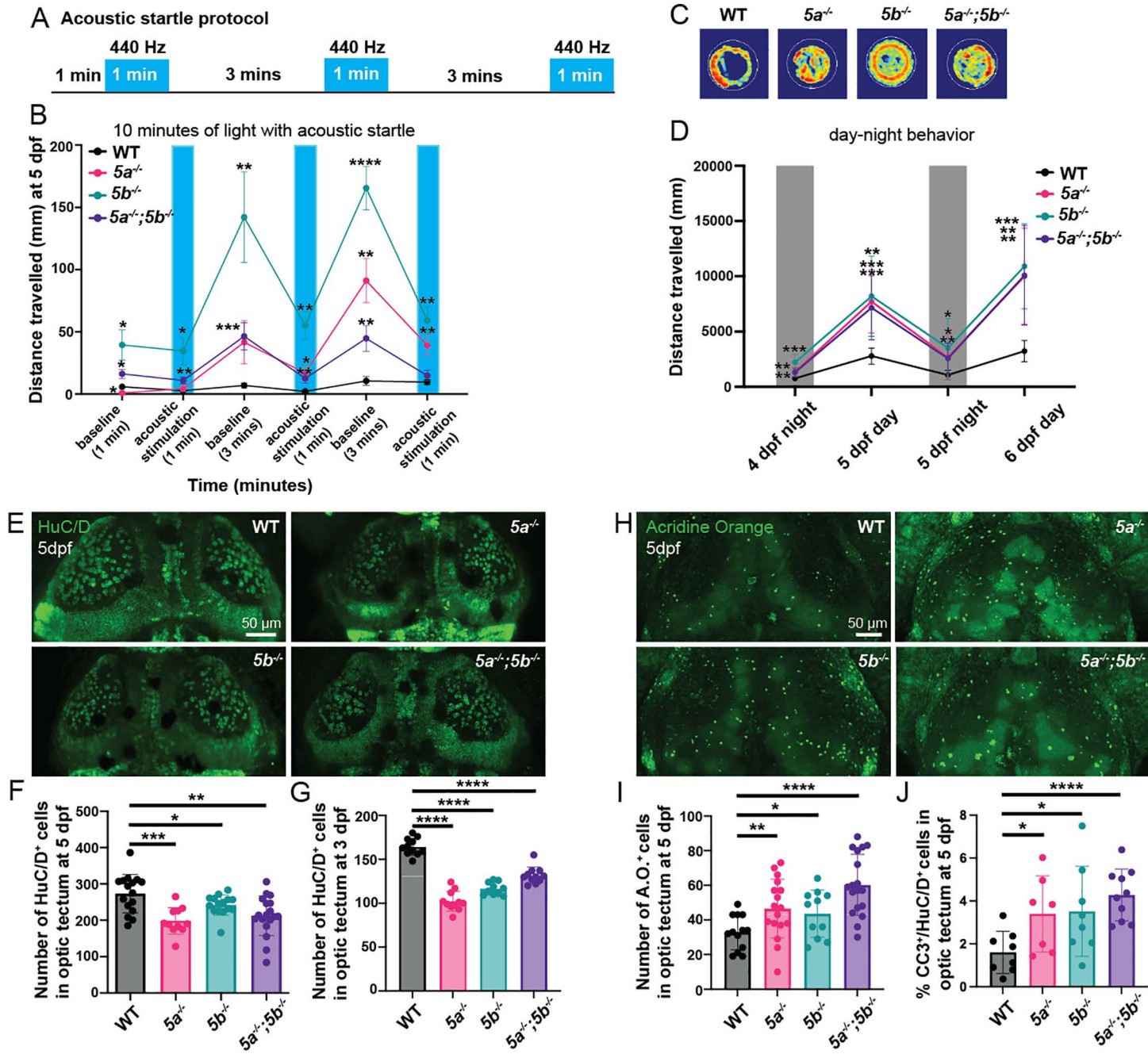

**Fig 2. Analysis of startle response and circadian disturbances in *slc13a5* mutants, along with their neuron population and neuronal apoptosis assessment.** (A) Schematic representation of acoustic startle protocol. (B) Quantification of distance traveled during baseline and acoustic stimulation periods at 5 dpf. *slc13a5* mutants are hyperactive in light without any external stimuli and are more responsive to startle, with significantly higher distance traveled during baseline and acoustic stimulation periods compared to WT. WT, n = 21; $5a^{-/-}$, n = 8; $5b^{-/-}$, n = 8; $5a^{-/-};5b^{-/-}$, n = 24. 'ns' values are not shown on the graph due to space constraints. (C) Heat maps of locomotion of larvae at 5 dpf under acoustic startle, showing that *slc13a5* mutants are more active than WTs (red shows highest presence and blue shows least presence). WT, n = 4; $5a^{-/-}$, n = 4; $5b^{-/-}$, n = 4; $5a^{-/-};5b^{-/-}$, n = 4. (D) Quantification of distance traveled in 10 hours of darkness in night at 4 dpf, 6 hours of light in the daytime at 5 dpf, 10 hours of darkness in night at 5 dpf and 6 hours of light in the daytime at 6 dpf. *slc13a5* mutants are more active than WTs in light as well as in darkness. WT, n = 10; $5a^{-/-}$, n = 10; $5b^{-/-}$, n = 10; $5a^{-/-};5b^{-/-}$, n = 10 for each time point. (E) 5 dpf WTs and *slc13a5* mutants; α-HuC/D (green). (F) Quantification of HuC/D⁺ cells in the optic tectum at 5 dpf. *slc13a5* mutants show a reduction in neuron numbers compared to WTs. WT, n = 16; $5a^{-/-}$, n = 11; $5b^{-/-}$, n = 15; $5a^{-/-};5b^{-/-}$, n = 18. (G) Quantification of HuC/D⁺ cells in the optic tectum at 3 dpf. *slc13a5* mutants show a reduction in neuron numbers compared to WTs. WT, n = 11; $5a^{-/-}$, n = 11; $5b^{-/-}$, n = 10; $5a^{-/-};5b^{-/-}$,

n = 11. (H) Live 5 dpf *slc13a5* mutants and WTs, stained with A.O. (green). (I) Quantification of A.O.$^+$ cells in the optic tectum at 5 dpf. *slc13a5* mutants show an increase in A.O.$^+$ cell numbers compared to WTs. WT, n = 13; *5a*$^{-/-}$, n = 17; *5b*$^{-/-}$, n = 11; *5a*$^{-/-}$;*5b*$^{-/-}$, n = 18. (J) Percentage of CC3$^+$/HuC/D$^+$ cells (out of total HuC/D$^+$ cells) in the optic tectum at 5 dpf. *slc13a5* mutants show an increase in CC3$^+$ neuron population compared to WTs. WT, n = 8; *5a*$^{-/-}$, n = 7; *5b*$^{-/-}$, n = 8; *5a*$^{-/-}$;*5b*$^{-/-}$, n = 10. Data are Mean ± S.D., *P ≤ 0.05, **P ≤ 0.01, ***P ≤ 0.001, ****P ≤ 0.0001- Unpaired *t* test. A.O., acridine orange; CC3, cleaved-Caspase3. The data underlying this figure can be found in S1 Data.

## *slc13a5*$^{-/-}$ larvae experience excitation/inhibition imbalance, enhanced *fosab* expression, strong brain hyperexcitability, and compromised mitochondrial function

Next, in *slc13a5* mutant zebrafish we tested for disruption of neuronal excitatory/inhibitory (E/I) balance and synchronous hyperexcitation as found in mammalian [33,34] and zebrafish [23,35,36] epilepsies. Using qPCR at 5 dpf, we quantified excitatory glutamatergic (*vglut2a*) and inhibitory GABAergic (*gad1b*) neuronal marker expression in larval brains and showed that *vglut2a* transcripts were significantly upregulated, whereas *gad1b* transcripts were significantly downregulated in *slc13a5*$^{-/-}$ larvae compared to WTs, suggestive of an E/I imbalance (Fig 3A and 3B). Our group and others have shown previously that seizure induction in epilepsy animal models leads to an upregulated expression of immediate early genes (IEG) in the brain [20,23,37]. We used qPCR at 5 dpf to quantify the expression of an IEG, *fosab,* and observed a significant upregulation in *fosab* transcripts in the *slc13a5*$^{-/-}$ larvae compared to WTs, further suggestive of brain hyperexcitability (Fig 3C). This result was further validated by performing co-immunostaining for HuC/D and c-Fos (also known as Fosab), at 5 dpf where we observed a 193.9% increase in the percentage of Fosab$^+$ neurons in the optic tectum in *5a*$^{-/-}$;*5b*$^{-/-}$ mutants compared to WTs (Fig 3D and 3E).

Next, to characterize seizure-like events, we measured extracellular field potentials from the optic tectum of agarose-immobilized 6 dpf WT and *slc13a5*$^{-/-}$ larval brains. WTs showed no evidence of abnormal brain activity (Fig 3F); however, extracellular field recordings in 30% (*5a*$^{-/-}$), 10% (*5b*$^{-/-}$) and 7.4% (*5a*$^{-/-}$;*5b*$^{-/-}$) of *slc13a5*$^{-/-}$ larval brains revealed repetitive inter-ictal like discharges (<1s duration) with high-frequency, large-amplitude, above threshold (>0.2mV) spikes (1.7 ± 0.7 Hz; 0.5 ± 0.2 mV, mean ± SEM), consistent with a spontaneous epileptic phenotype (Figs S4A-E and 3G-I). Notably, 40% (*5a*$^{-/-}$), 50% (*5b*$^{-/-}$) and 18.5% (*5a*$^{-/-}$;*5b*$^{-/-}$) of *slc13a5*$^{-/-}$ larvae also displayed repetitive but below threshold (<0.2mV) brain activity (Figs 3I and S4E). No longer duration (>1 s) discharges characterized as ictal-like events [38] were identified in the mutants. Thus, to increase the sensitivity of the technique, we employed a 16-channel NeuroProbe to measure extracellular field potentials from the optic tectum of agarose-immobilized 5 dpf WT and *5a*$^{-/-}$;*5b*$^{-/-}$ larval brains. During the protocol, 91.7% WTs and 100% *5a*$^{-/-}$;*5b*$^{-/-}$ larvae occasionally twitched, producing large-amplitude (>0.2 mV) deflections in the recordings. These deflections were of short duration (<1 s), and thus they were easily distinguishable from epileptic activities. Besides these movement-induced deflections, WTs showed no evidence of abnormal brain activity (Fig 3J). In contrast, 16.7% *5a*$^{-/-}$;*5b*$^{-/-}$ larvae showed a distinct epileptic pattern, consisting of a series of short bursts, with a total duration of> 10s (Figs 3K and S5A-C), and 8.3% of *5a*$^{-/-}$;*5b*$^{-/-}$ larvae showed repetitive twitches (>5 s) at consistent frequency (~1.8 Hz) (Figs 3L and S5D), an indication of epileptic seizures [39]. Combined, these data support that our *slc13a5*$^{-/-}$ larvae exhibit seizure-like events, with the low percentage of *slc13a5* mutants showing such events (Fig 3M) potentially due to spatio-temporal restrictions of this electrophysiological approach.

Dysregulation of mitochondrial function is observed across various types of epilepsies, including those modeled in zebrafish [23,40–42], and here we used the Seahorse XF Flux Bioanalyzer to assess bioenergetics in our *slc13a5*$^{-/-}$ larvae at 6 dpf (Fig 4A). We observed a dampening in basal respiration (−20.1%), ATP-linked respiration (−36.6%), total mitochondrial respiration (−32.9%), and non-mitochondrial respiration (−6.1%) in *5a*$^{-/-}$;*5b*$^{-/-}$ larvae compared to WTs (Fig 4B-4E). In contrast, maximum respiratory capacity and reserve capacity were upregulated by 28.2% and 351.2%, respectively (*5a*$^{-/-}$;*5b*$^{-/-}$), whereas proton leaks were unchanged (Fig 4F-4H), consistent with other reports whereby the loss of citrate uptake by cells due to *SLC13A5* mutations causes changes in cellular metabolism [43–46].

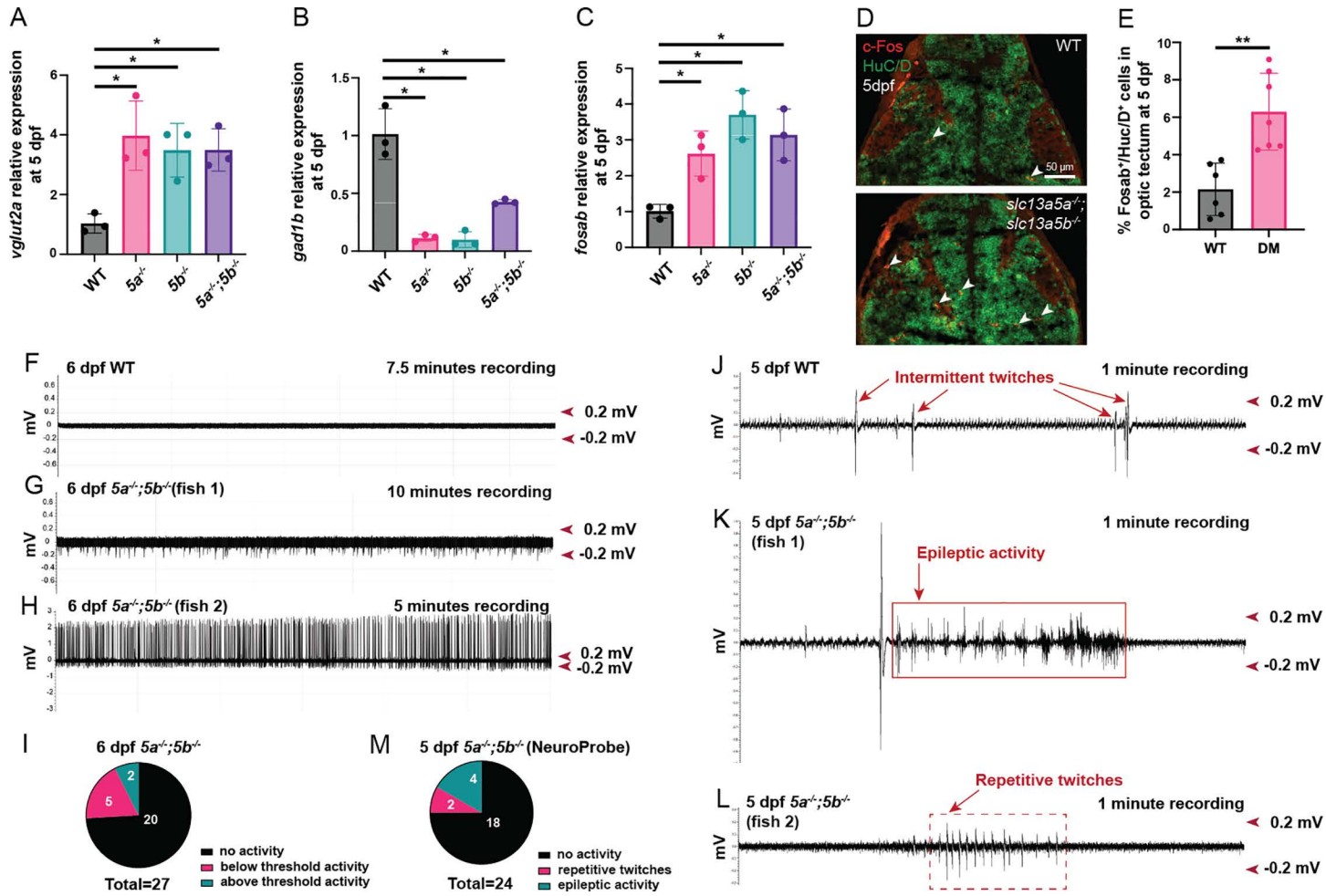

**Fig 3. E/I imbalance, *fosab* expression and brain hyperexcitability analysis in *slc13a5* mutants. (A-C)** qPCR analysis for relative *vglut2a*, *gad1b* and *fosab* mRNA expression in the heads of 5 dpf *slc13a5* mutants compared to WT. WT and *slc13a5* mutants, n = 3 × 10 larvae pooled. *vglut2a* is upregulated and *gad1b* is downregulated in *slc13a5* mutants indicating dysfunctional E/I balance. *fosab* is upregulated in *slc13a5* mutants indicating brain hyperexcitability. **(D, E)** 5 dpf WTs and *slc13a5* mutants; α-HuC/D (green), c-Fos (red). Percentage of Fosab+/HuC/D+ cells (out of total HuC/D+ cells) in the optic tectum at 5 dpf. *slc13a5* mutants show an increase in Fosab+ neuron population compared to WTs. WT, n=6; *5a−/−;5b−/−*, n = 7. **(F-I)** Representative extracellular recordings obtained from optic tectum of 6 dpf WTs and *slc13a5* mutants, and a pie chart of the number of *slc13a5* mutants showing different patterns of extracellular field potentials. The repetitive inter-ictal like discharges (<1s duration) with above threshold (>0.2mV), high-frequency, large-amplitude spikes seen in 7.4% *5a−/−;5b−/−* larvae are indicative of increased network hyperexcitability. 18.5% *5a−/−;5b−/−* larvae also displayed below threshold (<0.2mV) brain activity. WT, n=8 out of 8 with no abnormal activity. **(J-M)** Representative extracellular recordings obtained using a NeuroProbe placed in the optic tectum of 5 dpf WTs and *5a−/−;5b−/−* larvae, and a pie chart of number of *5a−/−;5b−/−* larvae showing different patterns of neuronal activity. 91.7% WTs and 100% *5a−/−;5b−/−* larvae intermittently twitched (as indicated by red arrows in J) but the deflections were distinguishable from epileptic activities. 16.7% *5a−/−;5b−/−* larvae showed epileptic pattern consisting of a series of short bursts, with a total duration of> 10s (as shown by red arrows in K) and 8.3% *5a−/−;5b−/−* larvae showed repetitive twitches (>5s) at consistent frequency of ~ 1.8 Hz (as shown by red arrows in L), another indication of seizures. WT, n=12 out of 12 with no abnormal activity. Data are Mean ± S.D., *P ≤ 0.05, **P ≤ 0.01- Unpaired *t* test. The data underlying this figure can be found in S1 Data.

## Transcriptomics analysis reveals differentially regulated pathways in *slc13a5−/−* larvae

Further, to unveil the transcriptomics signature of *slc13a5* mutants, we performed bulk RNA sequencing on pooled 5 dpf WT, *5a−/−*, *5b−/−* or *5a−/−;5b−/−* heads. Interestingly, principal component analysis (PCA) revealed that *5a−/−* and *5a−/−;5b−/−* larvae are more transcriptomically similar than *5b−/−* larvae and WTs (S6 Fig). Further, we observed that 400 genes were

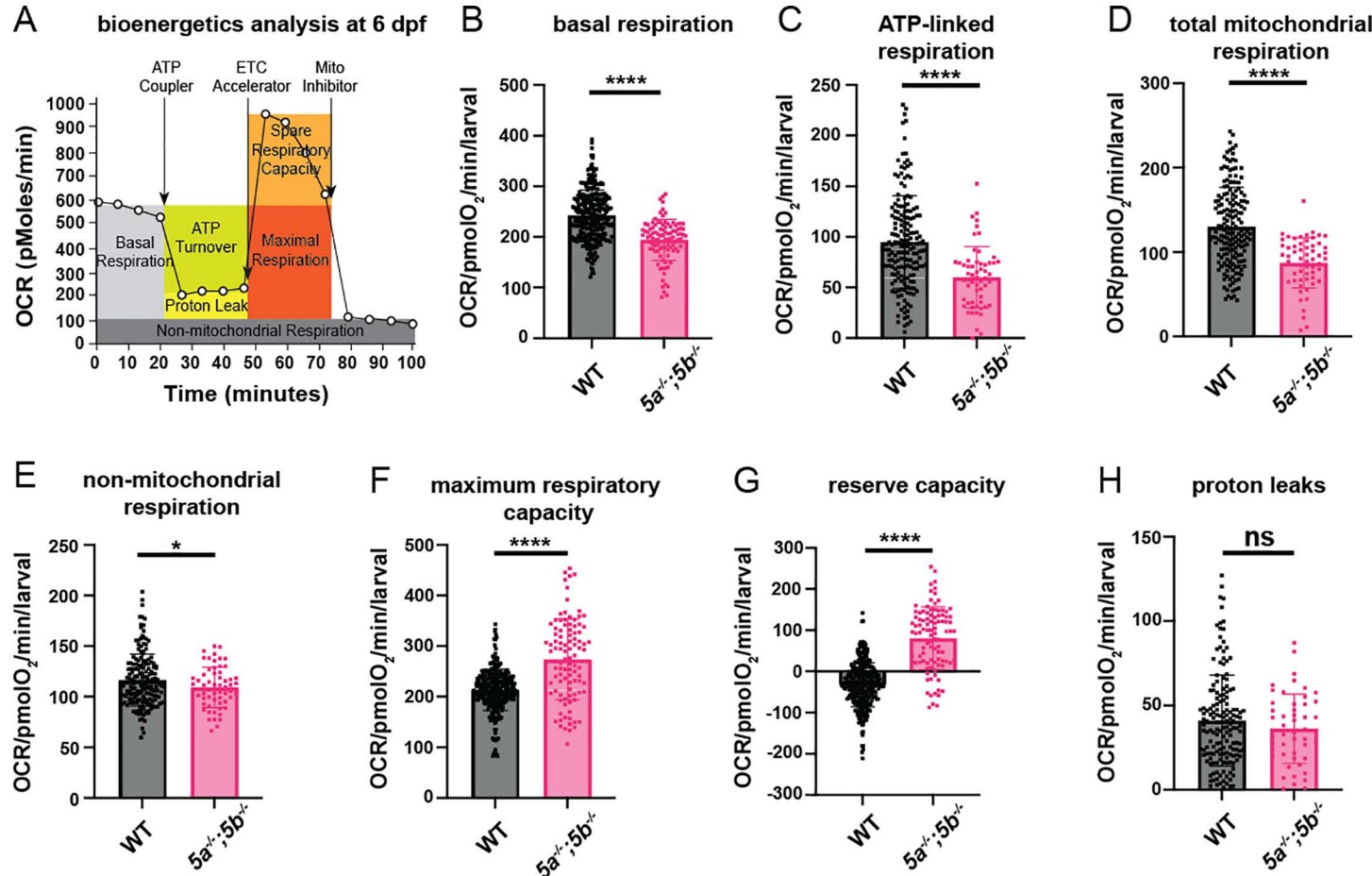

**Fig 4. Metabolic health analysis in *slc13a5* mutants.** (A) Schematic representation of how the Seahorse bioanalyzer displays mitochondrial bioenergetics being regulated by pharmacological inhibitors. (B) Quantification of basal respiration at 6 dpf. *slc13a5* mutants exhibit a significant reduction in basal respiration compared to WT. WT, n = 58; *slc13a5* mutants, n = 20 (individual values plotted from five cycles). (C) Quantification of ATP-linked respiration at 6 dpf. *slc13a5* mutants exhibit a significant reduction in ATP-linked respiration compared to WT. WT, n = 59; *slc13a5* mutants, n = 20 (individual values plotted from three cycles). (D) Quantification of total mitochondrial respiration at 6 dpf. *slc13a5* mutants exhibit a significant reduction in total mitochondrial respiration compared to WT. WT, n = 59; *slc13a5* mutants, n = 20 (individual values plotted from three cycles). (E) Quantification of non-mitochondrial respiration at 6 dpf. *slc13a5* mutants exhibit a significant reduction in non-mitochondrial respiration compared to WT. WT, n = 59; *slc13a5* mutants, n = 20 (individual values plotted from three cycles). (F) Quantification of maximum respiratory capacity at 6 dpf. *slc13a5* mutants exhibit a significant increase in maximum respiratory capacity compared to WT. WT, n = 59; *slc13a5* mutants, n = 20 (individual values plotted from five cycles). (G) Quantification of reserve capacity at 6 dpf. *slc13a5* mutants exhibit a significant increase in reserve capacity compared to WT. WT, n = 59; *slc13a5* mutants, n = 20 (individual values plotted from five cycles). (H) Quantification of proton leaks at 6 dpf. This parameter was unchanged in *slc13a5* mutants compared to WT. WT, n = 32; *slc13a5* mutants, n = 10 (individual values plotted from five cycles). Data are Mean ± S.D., ns: no significant changes observed, *P ≤ 0.05, ****P ≤ 0.0001- Unpaired *t* test. The data underlying this figure can be found in S1 Data.

significantly upregulated and 380 genes were significantly downregulated (padj < 0.05 and log2FC > 1.5 or < −1.5) in *5a*[−/−] and *5a*[−/−]*;5b*[−/−] larvae compared to WTs (S7 and S8 Figs). In particular, we found genes involved in apoptosis and metabolism to be upregulated in *5a*[−/−] and *5a*[−/−]*;5b*[−/−] larvae (Fig 5A), validating our data showing increased neuronal death (Figs 2J and S3H) and disrupted neurometabolism (Fig 4). Additionally, we found genes involved in inflammation and immune response to be upregulated in *5a*[−/−] and *5a*[−/−]*;5b*[−/−] larvae (Fig 5A), indicative of possible neuroinflammation in these mutants due to seizure-induced neuronal apoptosis, as shown by others [47,48]. Our *5a*[−/−] and *5a*[−/−]*;5b*[−/−] larvae further showed downregulation of genes affecting circadian rhythm (Fig 5B), consistent with circadian behavioral deficits in these

## A Upregulated genes in 5a<sup>-/-</sup>, 5a<sup>-/-</sup>;5b<sup>-/-</sup> and 5b<sup>-/-</sup> compared to WT

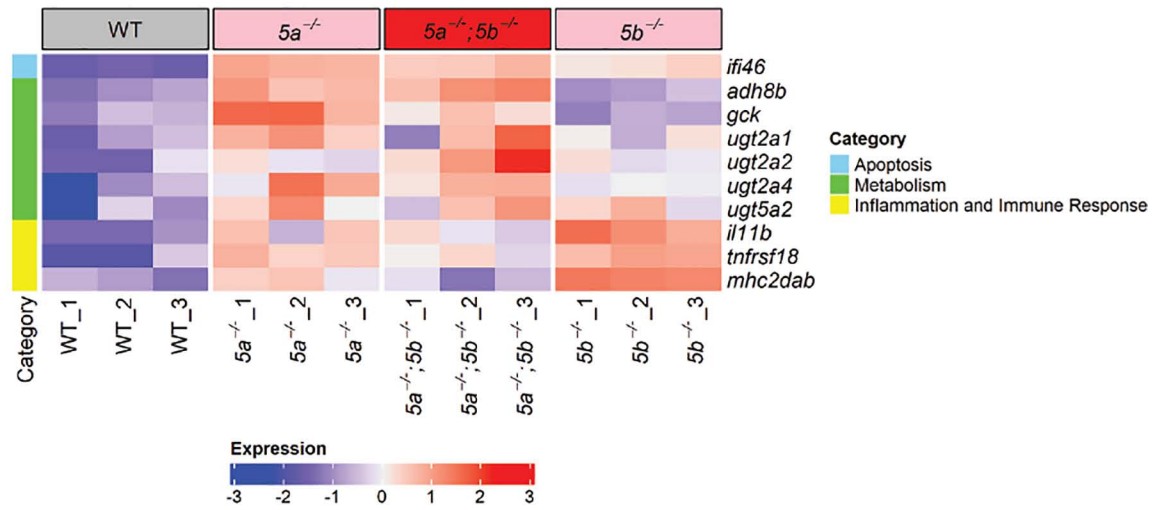

## B Downregulated genes in 5a<sup>-/-</sup>, 5a<sup>-/-</sup>;5b<sup>-/-</sup> and 5b<sup>-/-</sup> compared to WT

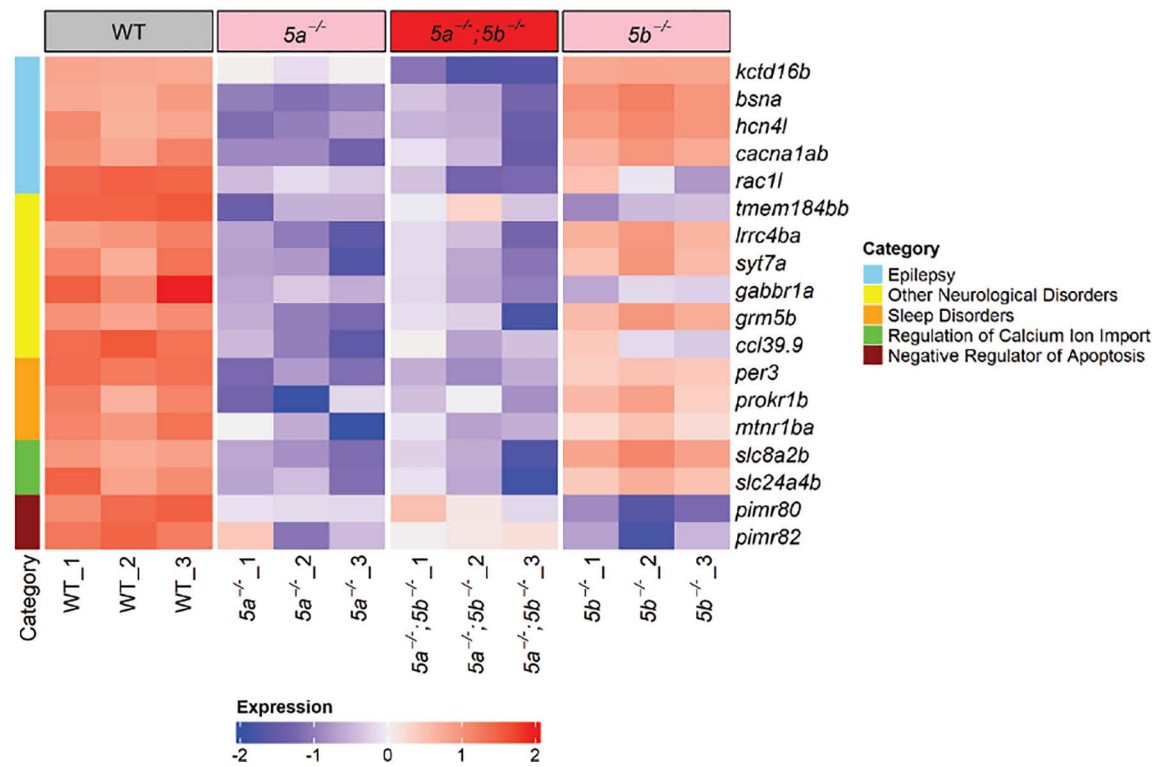

**Fig 5. Transcriptomics profiling of *slc13a5* mutants. (A**-B) Bulk RNA sequencing was performed on pools of zebrafish heads of 5 dpf WT, *5a<sup>-/-</sup>*, *5b<sup>-/-</sup>* and *5a<sup>-/-</sup>;5b<sup>-/-</sup>*. WT, *5a<sup>-/-</sup>*, *5b<sup>-/-</sup>* and *5a<sup>-/-</sup>;5b<sup>-/-</sup>*, n = 3 × 15 heads pooled. Heatmaps were constructed for prioritized DEGs using ComplexHeatmap. Genes upregulated in *5a<sup>-/-</sup>* and *5a<sup>-/-</sup>;5b<sup>-/-</sup>* larvae compared to WTs (padj < 0.05 and log2FC > 1.5) are involved in apoptosis, inflammation and immune

response and metabolism (A). Genes upregulated in *5b⁻/⁻* larvae compared to WTs (padj < 0.05 and log2FC > 1.5) are involved in apoptosis and inflammation and immune response (A). Genes downregulated in *5a⁻/⁻* and *5a⁻/⁻;5b⁻/⁻* larvae compared to WTs (padj < 0.05 and log2FC < −1.5) are involved in epilepsy and other neurological disorders, calcium ion transport, sleep regulation and negative regulation of apoptosis (B). Genes downregulated in *5b⁻/⁻* larvae compared to WTs (padj < 0.05 and log2FC < −1.5) are involved in neurological disorders and negatively regulate apoptosis (B). Heatmaps with a full list of DEGs are provided in S7–S10 Figs. DEGs, differentially expressed genes. The data underlying this figure can be found in S2 Data.

mutants (Fig 2D). Some genes whose loss of function is associated with epilepsy or other neurological disorders also were downregulated, as were genes regulating calcium ion transport (Fig 5B), pointing towards disturbed calcium homeostasis as a plausible cause of seizures in these mutants. Also, genes involved in the negative regulation of apoptosis were downregulated (Fig 5B), further validating previously shown enhanced apoptosis in these mutants (Figs 2J and S3H).

Next, we observed that 304 genes were significantly upregulated and 444 genes were significantly downregulated (padj < 0.05 and log2FC > 1.5 or < −1.5) in *5b⁻/⁻* larvae compared to WTs (S9 and S10 Figs). Specifically, genes involved in inflammation and immune response were upregulated in *5b⁻/⁻* larvae (Fig 5A), indicative of activated neuroinflammatory pathway in these mutants perhaps due to seizure-induced neuronal apoptosis (Figs 2J and S3H). Further, genes involved in apoptosis were upregulated (Fig 5A) and those that negatively regulate apoptosis were downregulated (Fig 5B) in *5b⁻/⁻* larvae, consistent with our data showing increased neuronal death in these mutants (Figs 2J and S3H). Some genes whose loss of function is associated with neurological disorders were also downregulated (Fig 5B). Finally, these studies revealed that *5a⁻/⁻* and *5a⁻/⁻;5b⁻/⁻* larvae exhibit similar transcriptomic signatures and thus, we focused our further study on *5a⁻/⁻;5b⁻/⁻* larvae.

## A mechanistic link between high extracellular citrate and seizure susceptibility in *slc13a5⁻/⁻* larvae

The underlying mechanism linking citrate transporter mutations to seizures remains debated [11], with one hypothesis being that the accumulation of extracellular citrate chelates the ions important for regulating neuronal membrane depolarizations and E/I balance. N-methyl-D-aspartate (NMDA) receptors are glutamate-gated channels that allow the timely influx of calcium ($Ca^{+2}$) to mediate excitatory signaling. Zinc is a critical NMDA receptor inhibitor that helps regulate $Ca^{+2}$ uptake and prevent prolonged excitation. Citrate is a potent zinc chelator, which when present in excess might bind all the zinc and prevent it from modulating NMDA receptor function, causing unremitting $Ca^{+2}$ influx and excitatory signaling.

First, we tested whether *slc13a5* mutants display increased $Ca^{+2}$ events, as would be predicted if NMDA receptor signaling was overactive. We crossed WT and *5a⁻/⁻;5b⁻/⁻* larvae with the *Tg(elavl3:Hsa.H2B-GCaMP6s)* zebrafish that express a neuron-specific calcium biosensor, and performed live calcium imaging at 3 dpf. *5a⁻/⁻;5b⁻/⁻* larvae displayed a 19% increase in calcium amplitude and 469% in the frequency of calcium events compared to WTs (Fig 6A and 6B, S1 and S2 Movies). The representative single neuron calcium traces illustrate the extent of increased $Ca^{+2}$ signaling in the *5a⁻/⁻;5b⁻/⁻* mutants (Figs 6C and S11A-F). Next, we assessed phosphorylated ERK (pERK) levels, an indirect downstream reporter of calcium signaling. We co-stained with pERK- and total ERK- (tERK) selective antibodies and imaged 5 dpf larval brains. Using a standard zebrafish reference brain to calculate normalized pERK levels as described previously [49,50], whole-brain activity maps were generated with voxels exhibiting significantly higher (green) and lower (magenta) pERK levels in the different brain regions. We found a significant increase in pERK signaling (green) in the FB, MB, and HB in *5a⁻/⁻;5b⁻/⁻* mutants compared to WTs, signifying upregulation of ERK phosphorylation possibly due to enhanced calcium influx in the neurons (Fig 6D). Notably, the lower pERK intensity observed in the eyes (magenta) is likely due to autofluorescence while imaging and therefore we do not consider it further. Owing to an increased pERK signaling in the FB and HB of *5a⁻/⁻;5b⁻/⁻* mutants, we performed live calcium imaging at 3 dpf (as per above) in these brain regions as well. *5a⁻/⁻;5b⁻/⁻* larvae displayed an increase of 8.3% in the amplitude and 82.3% in the frequency of calcium events compared to WTs in the FB (S11G and S11H Fig, S3 and S4 Movies), with representative single neuron calcium traces showing the extent of increased $Ca^{+2}$ signaling in the *5a⁻/⁻;5b⁻/⁻* mutants (S11I Fig). Interestingly, *5a⁻/⁻;5b⁻/⁻* larvae displayed an

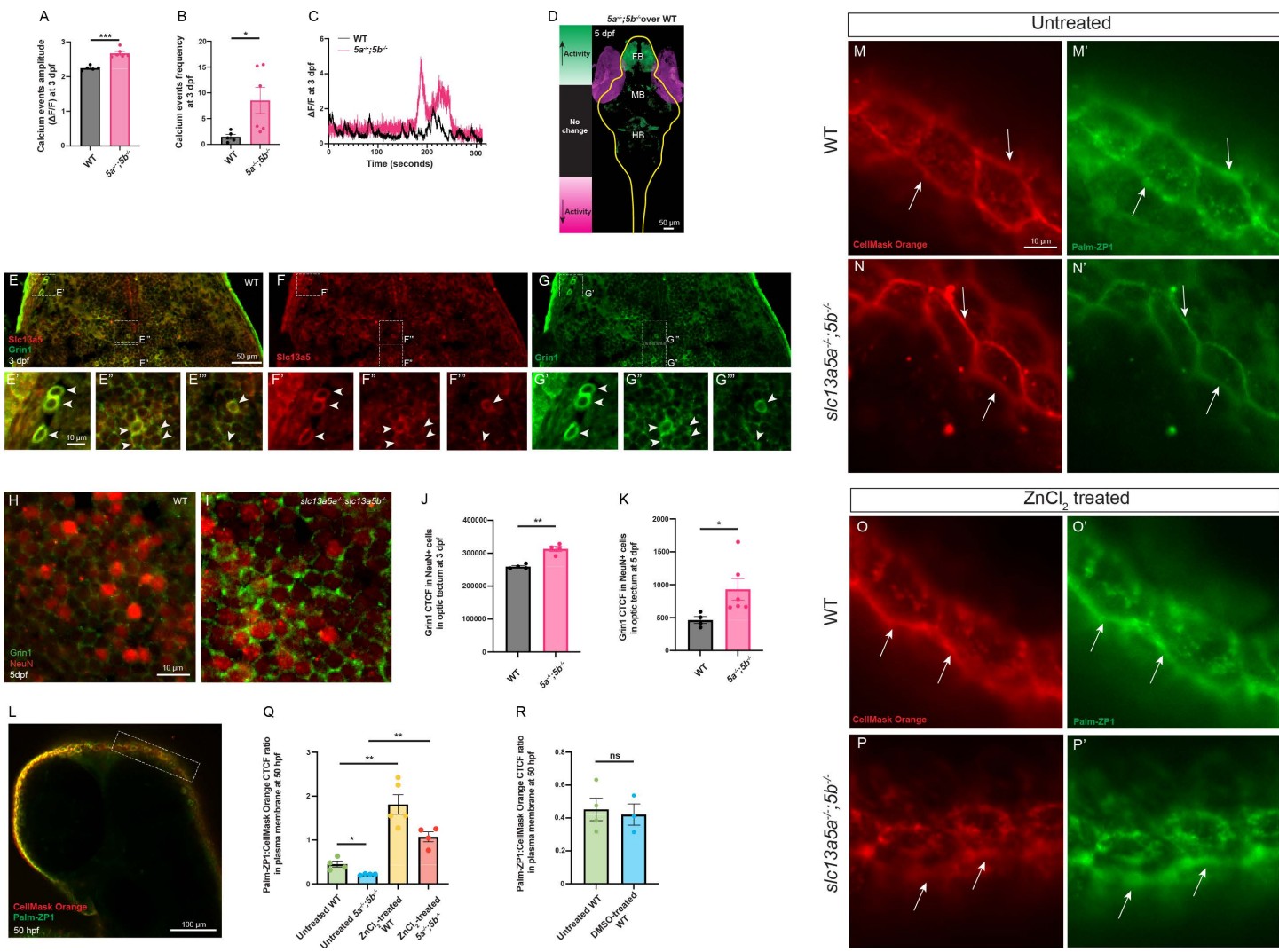

**Fig 6. Assessment of calcium events, pERK levels, NMDA receptor expression and extracellular zinc levels in *slc13a5* mutants. (A-**C) Quantification of the amplitude and frequency of calcium events (ΔF/F > 2) and representative single neuron calcium traces in the MB at 3 dpf. *5a⁻/⁻;5b⁻/⁻* larvae show a significant increase in calcium events compared to WTs. WT, n = 5; *5a⁻/⁻;5b⁻/⁻*, n = 6. (D) Whole-brain activity MAP-map depicting significant changes in pERK signal, calculated using Mann-Whitney U statistic Z score. The significance threshold was set based on an FDR whereby 0.05% of control pixels is set as significant. Voxels exhibiting significantly higher intensity values of pERK are denoted in green and those exhibiting significantly lower pERK intensity values are depicted in magenta, in the *slc13a5* mutants compared to WTs at 5 dpf. *slc13a5* mutants show an enhanced neural activity via increased pERK levels in different regions of the brain (green) compared to WT. The lower pERK intensity shown in the eyes (magenta) could be a background autofluorescence captured while imaging. WT, n = 9; *5a⁻/⁻;5b⁻/⁻*, n = 8. (E) Section of 3 dpf MB region of WTs; α-Slc13a5, α-Grin1. **(E', E", E''')** Higher magnification of dashed boxes in E. White arrowheads point to Slc13a5⁺/Grin1⁺ cells in MB. **(F, F', F", F''')** Slc13a5 expression shown in the brain section from E. (G, G', G", G''') Grin1 expression shown in the brain section for E. (H, I) 5 dpf WTs and *5a⁻/⁻;5b⁻/⁻* larvae; Grin1 (green), NeuN (red). **(J,** K) Quantification of the fluorescence intensity (CTCF) of Grin1 (α-Grin1) in the neurons (NeuN⁺) of optic tectum. *5a⁻/⁻;5b⁻/⁻* larvae show a significant increase in Grin1 expression in the neurons of optic tectum compared to WTs at 3 dpf and 5 dpf. WT, n = 4; *5a⁻/⁻;5b⁻/⁻*, n = 4 at 3 dpf. WT, n = 4; *5a⁻/⁻;5b⁻/⁻*, n = 6 at 5 dpf. (L) 50 hpf larvae stained with CellMask Orange and Palm-ZP1. **(M, M', N, N',** Q) Quantification of the fluorescence intensity (CTCF) ratios of Palm-ZP1:CellMask Orange in the plasma membrane. *5a⁻/⁻;5b⁻/⁻* larvae show a significant decrease in Palm-ZP1:CellMask Orange CTCF ratio in the plasma membrane compared to WTs. Untreated WT, n = 4; Untreated *5a⁻/⁻;5b⁻/⁻*, n = 4. White arrows indicate the plasma membrane regions used for CTCF measurements. **(O, O', P, P',** Q) Quantification of the fluorescence intensity (CTCF) ratios of Palm-ZP1:CellMask Orange in the plasma membrane. ZnCl₂-treated WTs show a significant increase in Palm-ZP1:CellMask Orange CTCF ratio in the plasma membrane compared to untreated WTs. Untreated WT, n = 4; ZnCl₂-treated WT, n = 5. Similarly, ZnCl₂-treated *5a⁻/⁻;5b⁻/⁻* larvae show a significant increase in Palm-ZP1:CellMask Orange CTCF ratio in the plasma membrane compared to untreated *5a⁻/⁻;5b⁻/⁻* larvae. Untreated *5a⁻/⁻;5b⁻/⁻*, n = 4; ZnCl₂-treated *5a⁻/⁻;5b⁻/⁻*, n = 4. White arrows indicate the plasma membrane regions used for CTCF measurements. (R) Untreated WTs (i.e., in E3 water) and DMSO-treated WTs (vehicle for Palm-ZP1) showed no significant changes in Palm-ZP1:CellMask Orange CTCF ratios in the plasma membrane. Untreated WTs, n = 4; DMSO-treated

WTs, n = 3. Data are Mean ± S.E.M., ns: no significant changes observed, *P ≤ 0.05, **P ≤ 0.01, ***P ≤ 0.001- Unpaired *t* test. FDR, false discovery rate. CTCF, corrected total cell fluorescence. The data underlying this figure can be found in S1 Data.

increase of 258.5% in the frequency of calcium events with no significant change in the amplitude compared to WTs in the HB (S11J and S11K Fig, S5 and S6 Movies), with representative single neuron calcium traces depicting the extent of increased Ca$^{+2}$ signaling in the *5a$^{-/-}$;5b$^{-/-}$* mutants (S11L Fig).

Next, we wanted to determine the involvement of NMDA receptor signaling in *slc13a5* mutants. NMDA receptor subunit *grin1* (also known as *glun1*) is reported to be responsible for calcium permeability of NMDA receptors [51,52]. Both *grin1a* and *grin1b* paralogs are expressed in the zebrafish brain and their loss of function affects behaviors [53,54]. We confirmed Grin1 expression in Slc13a5$^+$ MB neurons by immunostaining 3 dpf larvae and observed their co-expression in several MB cells (Fig 6E-G). Further, we co-immunostained 3 dpf and 5 dpf larvae with Grin1 and NeuN (mature neuronal marker) and observed a significant increase in the Grin1 fluorescence intensity, as measured by calculating CTCF in the optic tectum of *5a$^{-/-}$;5b$^{-/-}$* larvae compared to WTs, indicating active NMDA receptor signaling in *slc13a5* mutant brains (Fig 6H-K).

To determine whether excessive extracellular citrate causes zinc chelation in the *slc13a5$^{-/-}$* larvae, we incorporated a fluorescent zinc sensor, Palm-ZP1 that detects zinc in the extracellular side of the plasma membrane [55]. Fifty hpf larvae were co-incubated in 5 μM Palm-ZP1 and 5 μg/ml CellMask Orange (plasma membrane marker dye) for 15 min at 28°C, followed by live imaging the brain region (Fig 6L). We observed a −53.3% reduction in the ratio of Palm-ZP1:Cell Mask Orange fluorescence intensity, as measured by calculating Palm-ZP1:Cell Mask Orange CTCF ratio in the plasma membrane of *5a$^{-/-}$;5b$^{-/-}$* larvae compared to WTs (Fig 6M, 6M', 6N, 6N' and 6Q), suggesting possible zinc chelation due to elevated extracellular citrate. Notably, between untreated WTs (e.g., in E3 water) and DMSO-treated WTs (vehicle for Palm-ZP1), there were no significant differences in the Palm-ZP1:CellMask Orange fluorescence intensity ratios, as measured by calculating CTCF ratios in the plasma membrane (Fig 6R). Further, prior incubation in 25 μM ZnCl$_2$ for 2 hours led to an increase of 302.2% in Palm-ZP1:CellMask Orange CTCF ratio in WTs compared to untreated WTs (Fig 6M, 6M', 6O, 6O' and 6Q). Similarly, ZnCl$_2$-treated *5a$^{-/-}$;5b$^{-/-}$* larvae also exhibited an increase of 409.5% in Palm-ZP1:CellMask Orange CTCF ratio compared to untreated mutants (Fig 6N, 6N', 6P, 6P' and 6Q).

Collectively, these data suggest that *slc13a5* mutants exhibit overactivated NMDA receptor signaling due to zinc chelation by high extracellular citrate.

## Suppression of NMDA signaling and zinc treatment rescue dysregulated calcium activity, neurometabolism, and startle behaviors in *slc13a5$^{-/-}$* larvae

Next, we used pharmacological and genetic approaches to investigate whether NMDA receptor signaling influenced *5a$^{-/-}$;5b$^{-/-}$* larvae phenotypes. Pharmacologically, we exposed WT and mutant zebrafish larvae to memantine, an established NMDA antagonist used in other zebrafish studies [56]. Genetically, we designed morpholinos to knock down both the paralogs of NMDA receptor subunit *grin1*. We co-injected *grin1a;grin1b* morpholinos (MOs) in the WT and *5a$^{-/-}$;5b$^{-/-}$* zebrafish, expressing the *Tg(elavl3:Hsa.H2B-GCaMP6s)* transgene and performed live calcium imaging in the MB at 3 dpf. Fifty μM memantine-treatment for 2 hours prior to imaging and *grin1a;grin1b* MO injection in *5a$^{-/-}$;5b$^{-/-}$* larvae caused a reduction of −17.5% (memantine-treated) and −18.6% (*grin1a;grin1b* MO injected) in the amplitude and −72.7% (memantine-treated) and −83.3% (*grin1a;grin1b* MO injected) in the frequency of calcium events compared to vehicle-treated (0.5% DMSO) or control MO-injected *5a$^{-/-}$;5b$^{-/-}$* larvae (Fig 7A and 7B, S7-S9 Movies). The representative single neuron calcium traces illustrated that *5a$^{-/-}$;5b$^{-/-}$* larvae exhibited fewer above-threshold events after memantine treatment or *grin1a;grin1b* MO injection (Fig 7C). Notably, there were no significant changes in the amplitude or frequency of calcium events in WT larvae after treatment with memantine or *grin1a;grin1b* MO injections (S12A and S12B Fig). Importantly, a significant reduction in the Grin1 fluorescence intensity, as measured by calculating CTCF in the brains of

PLOS Biology

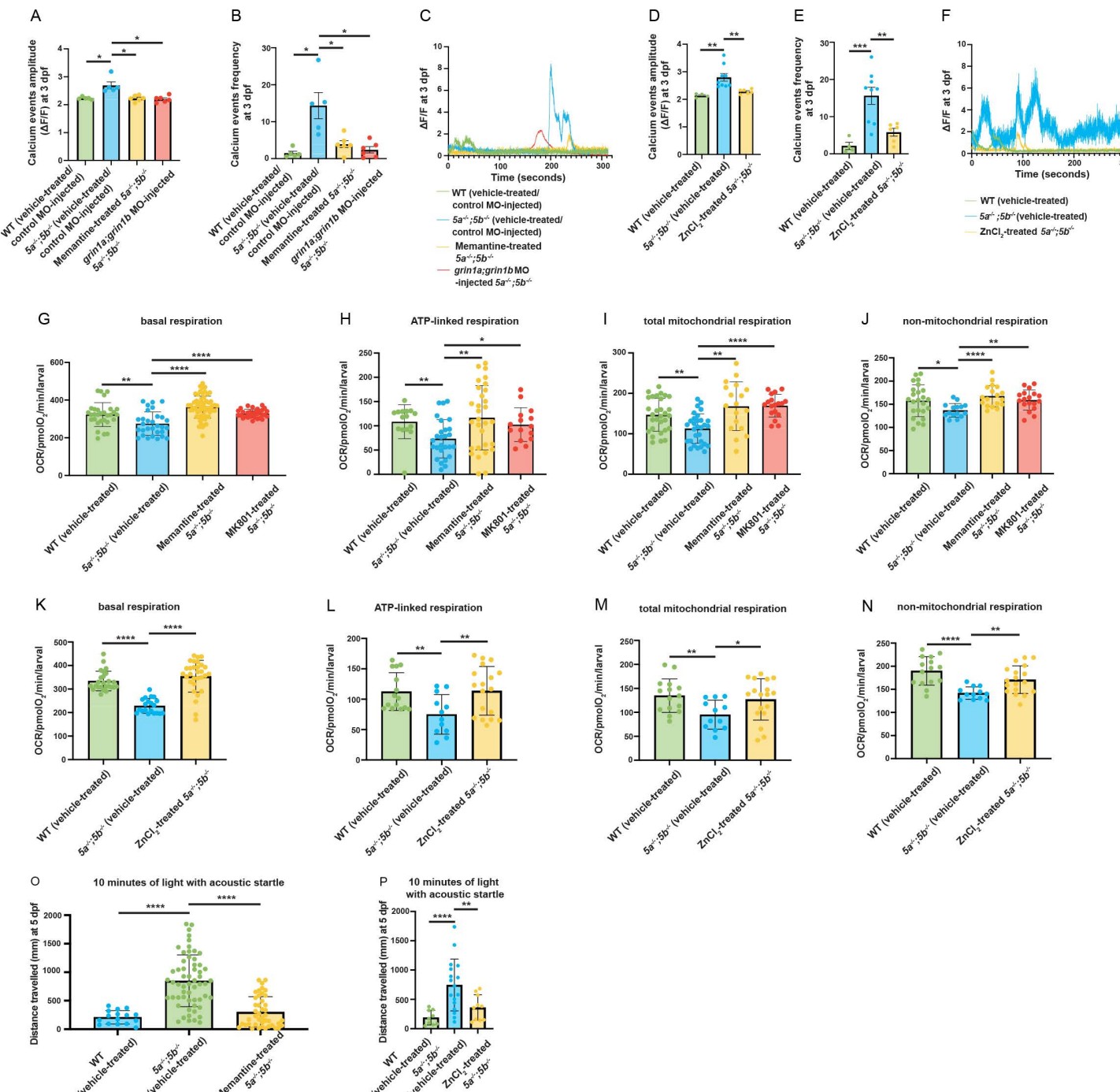

**Fig 7. Effect of NMDA receptor signaling suppression and zinc treatment on *slc13a5* mutants. (A-**C) Quantification of the amplitude and frequency of calcium events (ΔF/F > 2) and representative single neuron calcium traces in the MB at 3 dpf. Memantine treatment and *grin1a;grin1b* MO injections significantly reduced the amplitude and frequency of calcium events in *5a*⁻/⁻;*5b*⁻/⁻ larvae compared to vehicle-treated or control MO-injected *5a*⁻/⁻;*5b*⁻/⁻ larvae. Vehicle-treated/control MO-injected WT larvae, n = 5; vehicle-treated/control MO-injected *5a*⁻/⁻;*5b*⁻/⁻ larvae, n = 5; memantine-treated *5a*⁻/⁻;*5b*⁻/⁻ larvae, n = 6; *grin1a;grin1b* MO-injected *5a*⁻/⁻;*5b*⁻/⁻ larvae, n = 6. **(D-**F) Quantification of the amplitude and frequency of calcium events (ΔF/F > 2) and representative single neuron calcium traces in the MB at 3 dpf. ZnCl₂ treatment significantly reduced the amplitude and frequency of calcium events in *5a*⁻/⁻;*5b*⁻/⁻ larvae compared to vehicle-treated *5a*⁻/⁻;*5b*⁻/⁻ larvae. Vehicle-treated WT larvae, n = 4; vehicle-treated *5a*⁻/⁻;*5b*⁻/⁻ larvae, n = 9; ZnCl₂-treated *5a*⁻/⁻;*5b*⁻/⁻ larvae, n = 6. (G) Quantification of basal respiration before and after treatment with memantine and MK801 at 6 dpf. Memantine and MK801 treatments rescue compromised basal respiration in *5a*⁻/⁻;*5b*⁻/⁻ larvae. Vehicle-treated WT, n = 6; vehicle-treated *5a*⁻/⁻;*5b*⁻/⁻, n = 6; memantine-treated

5a$^{-/-}$;5b$^{-/-}$, n = 12; MK801-treated 5a$^{-/-}$;5b$^{-/-}$, n = 6 (individual values plotted from five cycles). (H) Quantification of ATP-linked respiration before and after treatment with memantine and MK801 at 6 dpf. Memantine and MK801 treatments rescue compromised ATP-linked respiration in 5a$^{-/-}$;5b$^{-/-}$ larvae. Vehicle-treated WT, n = 5; vehicle-treated 5a$^{-/-}$;5b$^{-/-}$, n = 9; memantine-treated 5a$^{-/-}$;5b$^{-/-}$, n = 10; MK801-treated 5a$^{-/-}$;5b$^{-/-}$, n = 5 (individual values plotted from three cycles). (I) Quantification of total mitochondrial respiration before and after treatment with memantine and MK801 at 6 dpf. Memantine and MK801 treatments rescue compromised total mitochondrial respiration in 5a$^{-/-}$;5b$^{-/-}$ larvae. Vehicle-treated WT, n = 10; vehicle-treated 5a$^{-/-}$;5b$^{-/-}$, n = 10; memantine-treated 5a$^{-/-}$;5b$^{-/-}$, n = 6; MK801-treated 5a$^{-/-}$;5b$^{-/-}$, n = 6 (individual values plotted from three cycles). (J) Quantification of non-mitochondrial respiration before and after treatment with memantine and MK801 at 6 dpf. Memantine and MK801 treatments rescue compromised non-mitochondrial respiration in 5a$^{-/-}$;5b$^{-/-}$ larvae. Vehicle-treated WT, n = 8; vehicle-treated 5a$^{-/-}$;5b$^{-/-}$, n = 5; memantine-treated 5a$^{-/-}$;5b$^{-/-}$, n = 6; MK801-treated 5a$^{-/-}$;5b$^{-/-}$, n = 6 (individual values plotted from three cycles). (K) Quantification of basal respiration before and after treatment with ZnCl$_2$ at 6 dpf. ZnCl$_2$ treatment rescues compromised basal respiration in 5a$^{-/-}$;5b$^{-/-}$ larvae. Vehicle-treated WT, n = 5; vehicle-treated 5a$^{-/-}$;5b$^{-/-}$, n = 4; ZnCl$_2$-treated 5a$^{-/-}$;5b$^{-/-}$, n = 6 (individual values plotted from five cycles). (L) Quantification of ATP-linked respiration before and after treatment with ZnCl$_2$ at 6 dpf. ZnCl$_2$ treatment rescues compromised ATP-linked respiration in 5a$^{-/-}$;5b$^{-/-}$ larvae. Vehicle-treated WT, n = 5; vehicle-treated 5a$^{-/-}$;5b$^{-/-}$, n = 4; ZnCl$_2$-treated 5a$^{-/-}$;5b$^{-/-}$, n = 6 (individual values plotted from three cycles). (M) Quantification of total mitochondrial respiration before and after treatment with ZnCl$_2$ at 6 dpf. ZnCl$_2$ treatment rescues compromised total mitochondrial respiration in 5a$^{-/-}$;5b$^{-/-}$ larvae. Vehicle-treated WT, n = 5; vehicle-treated 5a$^{-/-}$;5b$^{-/-}$, n = 4; ZnCl$_2$ -treated 5a$^{-/-}$;5b$^{-/-}$, n = 6 (individual values plotted from three cycles). (N) Quantification of non-mitochondrial respiration before and after treatment with ZnCl$_2$ at 6 dpf. ZnCl$_2$ treatment rescues compromised non-mitochondrial respiration in 5a$^{-/-}$;5b$^{-/-}$ larvae. Vehicle-treated WT, n = 5; vehicle-treated 5a$^{-/-}$;5b$^{-/-}$, n = 4; ZnCl$_2$ -treated 5a$^{-/-}$;5b$^{-/-}$, n = 6 (individual values plotted from three cycles). (O) Quantification of total distance traveled in 10 min in acoustic startle before and after treatment with memantine at 5 dpf. Memantine treatment rescues impaired startle response in 5a$^{-/-}$;5b$^{-/-}$ larvae. Vehicle-treated WT, n = 16; vehicle-treated 5a$^{-/-}$;5b$^{-/-}$, n = 58; memantine-treated 5a$^{-/-}$;5b$^{-/-}$, n = 43. (P) Quantification of total distance traveled in 10 min in acoustic startle before and after treatment with ZnCl$_2$ at 5 dpf. ZnCl$_2$ treatment rescues impaired startle response in 5a$^{-/-}$;5b$^{-/-}$ larvae. Vehicle-treated WT, n = 9; vehicle-treated 5a$^{-/-}$;5b$^{-/-}$, n = 17; memantine-treated 5a$^{-/-}$;5b$^{-/-}$, n = 10. Data are Mean ± S.E.M. and Mean ± S.D., *P ≤ 0.05, **P ≤ 0.01, ***P ≤ 0.001, ****P ≤ 0.0001-Unpaired t test. MO, morpholino. The data underlying this figure can be found in S1 Data.

Grin1 immunostained grin1a;grin1b MO-injected WTs compared to control MO-injected WTs at 3 dpf, was observed, confirming efficacy of grin1a;grin1b MOs (S12C Fig). These results demonstrate a role for NMDA receptors in slc13a5 mutant dysregulation of Ca$^{+2}$ signaling.

We further assessed the involvement of zinc chelation in 5a$^{-/-}$;5b$^{-/-}$ larvae phenotypes by using a pharmacological approach. We treated WT and 5a$^{-/-}$;5b$^{-/-}$ larvae, expressing the Tg(elavl3:Hsa.H2B-GCaMP6s) transgene, with 25 µM ZnCl$_2$ or vehicle (E3 water) for 2 hours, followed by live calcium imaging in the MB at 3 dpf. ZnCl$_2$ treatment in 5a$^{-/-}$;5b$^{-/-}$ larvae led to a reduction of −18.3% in the amplitude and −62.9% in the frequency of calcium events compared to vehicle-treated 5a$^{-/-}$;5b$^{-/-}$ larvae (Fig 7D and 7E, S10 and S11 Movies). The representative single neuron calcium traces illustrated that 5a$^{-/-}$;5b$^{-/-}$ larvae exhibited fewer above-threshold events after ZnCl$_2$ treatment (Fig 7F). Of note, we did not observe any significant changes in the amplitude or frequency of calcium events in WT larvae after treatment with ZnCl$_2$ (S12D and S12E Fig). These data support the role of zinc chelation in perturbed Ca$^{+2}$ signaling in slc13a5 mutants.

An alternative mechanistic hypothesis is that the loss of intracellular citrate disrupts neuronal homeostasis, leading to hyperexcitability. If this alternative hypothesis is true, then blocking NMDA receptors or treating with ZnCl$_2$ should have no additional effect in the mutants on a key measure of neuronal homeostasis: bioenergetics. 6 dpf 5a$^{-/-}$;5b$^{-/-}$ larvae were treated with 50 µM memantine, 50 µM MK801 (a second potent NMDA antagonist [54]) or vehicle (0.5% DMSO) for 2 hours prior to initiating the Seahorse assay (as per above). Blocking NMDA receptors significantly rescued several bioenergetics parameters dysregulated in the 5a$^{-/-}$;5b$^{-/-}$ larvae, including basal respiration, ATP-linked respiration, total mitochondrial respiration, and non-mitochondrial respiration (Fig 7G-7J). Further, 6 dpf 5a$^{-/-}$;5b$^{-/-}$ larvae were treated with 25 µM ZnCl$_2$ or vehicle (E3 water) for 2 hours, followed by performing Seahorse assay as described previously. ZnCl$_2$ treatment significantly rescued basal respiration, ATP-linked respiration, total mitochondrial respiration, and non-mitochondrial respiration in the 5a$^{-/-}$;5b$^{-/-}$ larvae (Fig 7K-7N). Combined, these data suggest that an interplay between extracellular citrate, zinc chelation and NMDA receptors mediates the seizure-like phenotype in slc13a5 mutants. Notably, memantine, MK801 and ZnCl$_2$ had no effect on these metabolic parameters in the WTs (S13A-H Fig).

Finally, to link NMDA receptor function and zinc chelation to circuit-level behaviors of slc13a5 mutant zebrafish, we examined whether the impaired startle behavior of our mutants was rescued by NMDA antagonism and ZnCl$_2$ treatment. We treated 5 dpf larvae with memantine (50 µM), ZnCl$_2$ (25 µM) or their respective vehicles for 2 hours, followed by the

acoustic startle protocol (Fig 2A). Treatment with memantine and ZnCl$_2$ rescued the impaired startle response in *5a^{-/-};5b^{-/-}* larvae, causing a significant reduction in the total distance swam during the complete protocol (Fig 7O, 7P). No effect was observed of memantine and ZnCl$_2$ exposure on the swimming activity of the WTs in this acoustic startle protocol; however, MK801 exposure significantly reduced the swimming activity of WTs, making it unsuitable for further analyses herein (S13I and S13J Fig).

Taken together, we propose a model whereby deficiencies in Slc13a5 cause accumulation of extracellular citrate that excessively chelates zinc, which fails to efficiently buffer NMDA receptor activity leading to increased Ca$^{+2}$ uptake inside the neurons and excessive excitatory firing that manifests as seizures. We further show that blocking NMDA receptors or treatment with zinc prevents enhanced cytoplasmic Ca$^{+2}$ influx into the neurons, thereby rescuing epileptic phenotypes in our zebrafish *slc13a5* mutants (Fig 8).

## Discussion

It remains unclear why children with mutations in *SLC13A5* develop severe epilepsy and associated comorbidities. In the present study, we created genetic loss-of-function mutants of both the paralogs of human *SLC13A5*, i.e., *5a* and *5b* in the zebrafish to facilitate the study of the underlying disease etiology. These *slc13a5* zebrafish mutants recapitulate many symptoms observed in *SLC13A5*-deficient individuals, making this zebrafish model a robust and reliable tool. Mechanistically, we show that 1) pharmacological inhibition or genetic knockdown of NMDA receptor signaling, and 2) treatment with zinc rescues the epileptic phenotypes, serving as the first empirical evidence for the 'chelation hypothesis' of SLC13A5 epilepsy. Combined, these zebrafish mutants are a translatable model to study underlying disease biology and may open new avenues of research toward novel treatment options for this disorder.

We first characterized the *slc13a5* mutants for disruption in various behaviors, which is a common strategy to test for gross neural network defects. Deficiencies in behaviors were observed across several assays in *slc13a5* mutants,

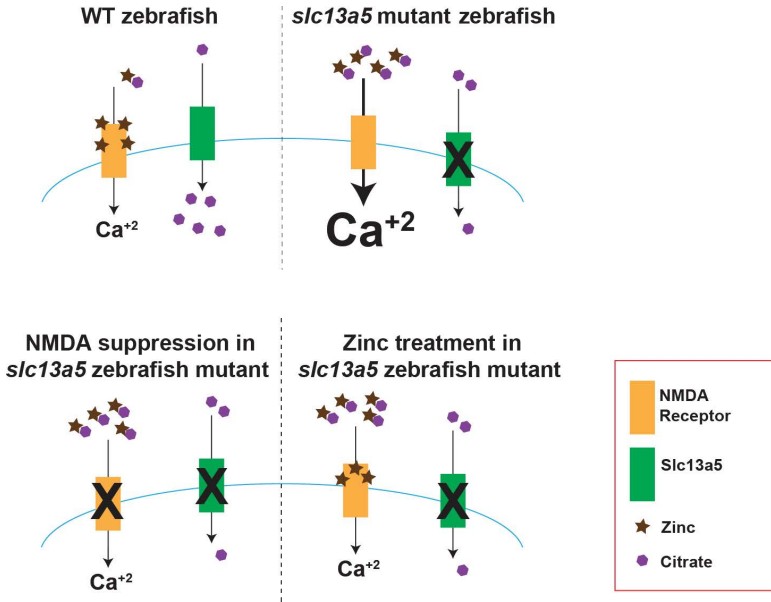

**Fig 8. Model of relationship between *slc13a5* mutations, NMDA receptor signaling and zinc chelation.** Unlike in WT larvae, *slc13a5* mutations in zebrafish presumably cause an increase in extracellular citrate levels that lead to zinc chelation, thereby preventing its inhibitory effect on NMDA receptors and ultimately leading to an excessive influx of calcium to the neurons, causing epilepsy. Blocking NMDA receptors and/or treatment with zinc prevents this enhanced influx of calcium into the neurons, thereby rescuing epilepsy-related phenotypes in the *slc13a5* mutants.

including those that recapitulate the sleep disturbances and perhaps cognitive disabilities (by showing impaired learning of the startle response) observed in *SLC13A5* individuals, further demonstrating the applicability of this model system. To investigate whether these behavioral phenotypes correlate with molecular changes in mutant brains, we focused our characterization on the optic tectum given the perhaps slightly stronger expression of *slc13a5* in the midbrain compared to other brain regions. There was a significant reduction in the number of neurons and midbrain size, along with a concomitant increase in the neuronal death in our *slc13a5* mutants. Microcephaly is reported in ~10% of SLC13A5 individuals [12,25], indicating that brain size reduction, possibly due to cell death as we report here, is present but is less penetrant in humans. We speculate that this increased apoptosis is a consequence of seizure-like events in the mutant brain, since other animal models and patient data show that epileptic seizures cause neuronal death via activation of apoptosis or autophagy signaling pathways [35,57–59]. It is also possible that a block in citrate transport causes a loss of intracellular citrate, a crucial metabolite for energy production, and thus alters neuronal metabolic health [44]. Our zebrafish *slc13a5* mutants exhibited compromised neurometabolism, consistent with a metabolic phenotype. Finally, it is notable that only a small percentage of the zebrafish mutants survive to adulthood, suggesting that perhaps disease progression becomes incompatible with survival in the zebrafish.

Brain MRI and EEG recordings from SLC13A5 epilepsy patients show punctate white matter lesions and abnormal brain activity, respectively [60,61]. Here, the local field potential electrophysiological recordings, as well as the upregulation of the immediate early gene, *fosab,* and perturbed E/I balance, is consistent with an epilepsy phenotype in our *slc13a5* mutants. This demonstration of seizure-like activity is not trivial given that citrate transporter mutations in *C. elegans, Drosophila,* and mice fail to robustly produce epileptiform events. That said, based on the combined data from our two electrophysiological assays, nearly 84% of the zebrafish $5a^{-/-};5b^{-/-}$ mutants, 70% of $5a^{-/-}$ mutants and 90% of $5b^{-/-}$ mutants displayed no hyperexcitability. One of the possible reasons could be an incomplete penetrance of seizure-like phenotype in these mutants at larval stages. The observation that only 5%–15% of these mutants survive to adulthood supports the idea that perhaps seizures in their brain emerge more frequently with age, leading to a high mortality rate. Nevertheless, based on a wide range of strong phenotypes displayed by our *slc13a5* mutants, including behavioral deficits, molecular defects, disturbed bioenergetics profile and importantly enhanced calcium events (another measure of neuronal activity), we can suggest that these mutants are a promising model of SLC13A5 epilepsy. Our transcriptomics profiling data support these phenotypes by showing that the genes involved in apoptosis, inflammation, metabolism and sleep are dysregulated in our *slc13a5* mutants. The RNA sequencing also reveals that overall, the $5a^{-/-}$ and $5a^{-/-};5b^{-/-}$ mutants exhibit similar transcriptional signatures that are distinct from $5b^{-/-}$ mutants, which perhaps isn't surprising given the varying degree of epileptic phenotypes amongst these three mutant categories.

A key finding of our studies is the mechanistic evidence suggesting an interplay between a defective citrate transporter, zinc chelation and activation of NMDA receptors. Little is known about the molecular etiology of this disorder and one of the hypotheses is that excessive extracellular citrate levels due to the loss of transmembrane transport chelates free zinc and prevents its binding to NMDA receptor subunits, thereby causing an uncontrolled influx of calcium into the neurons and leading to increased seizure susceptibility in the patients [11]. Our *slc13a5* mutants exhibit enhanced expression of NMDA receptors in the brain and a concomitant reduction in the zinc levels in the plasma membrane, both in support of this hypothesis. We also show that the *slc13a5* mutants exhibit upregulated levels of pERK, a downstream reporter of calcium activity, as well as a significant increase in calcium events as revealed by calcium imaging in live larvae. Interestingly, both NMDA signaling suppression (via memantine treatment and *grin1a;grin1b* MO injections) as well as zinc treatment led to a significant normalization of the amplitude and frequency of calcium events in the *slc13a5* mutants, demonstrating a role for zinc and NMDA receptors in *slc13a5* hyperexcitable phenotypes. We also show that pharmacological inhibition of NMDA signaling using memantine and MK801, and incubation with $ZnCl_2$ improves the metabolic profiles of our mutants, further validating a possible relationship between extracellular citrate, zinc and NMDA receptors. Further, memantine and zinc treatment rescued the impaired acoustic startle response of our mutant zebrafish,

demonstrating a functional interaction can rescue animal behaviors as well. Combined, these findings suggest that inhibition of NMDA receptor signaling and/or zinc treatment could be a potential therapeutic option for the treatment of epileptic seizures in SLC13A5 individuals.

A potential limitation of these studies is the expression of Slc13a5 in the plasma membrane of both neurons and astrocytes in zebrafish. In the human brain, SLC13A5 is expressed primarily in neurons [1,2,11], whereas in mice it is expressed in both neurons and astrocytes [62–64], which was considered a confounding factor when translating back to humans. And now, in zebrafish we show that Slc13a5 also is expressed in both neurons and astrocytes, albeit at higher levels in neurons, perhaps reflecting a functional difference in metabolic requirements for citrate by neural cells across vertebrate species. To circumvent any issues related to the dual expression of this citrate transporter in SLC13A5 epilepsy and to overcome any potential shortcomings of using zebrafish in isolation [65], induced pluripotent stem cells can be established from SLC13A5 individuals and used to generate human brain organoids. These brain organoids can be employed to validate these zebrafish findings in a human model and to overexpress *SLC13A5* to further shed insights into the function of SLC13A5 in the brain.

In conclusion, *slc13a5* mutant zebrafish are a promising model for the discovery of disease biology and a new tool for high-throughput drug screening and the testing of candidate drugs for this disorder.

## Materials and methods

### Zebrafish maintenance

Adult zebrafish (TL and AB strains) were maintained at 28°C in a 14-hour light/10-hour dark cycle under standard aquaculture conditions, and fertilized eggs were collected via natural spawning. The animals were fed twice daily with Artemia. Zebrafish embryos and larvae were maintained in E3 water in a non-$CO_2$ incubator (VWR) at 28°C on the same light–dark cycle as the aquatic facility. All protocols and procedures were approved by the Health Science Animal Care Committee (protocol number AC22–0153) at the University of Calgary in compliance with the Guidelines of the Canadian Council of Animal Care.

### Generation of zebrafish mutants by CRISPR/Cas9

*5a*$^{-/-}$ (*slc13a5a*$^{ca204/+}$) and *5b*$^{-/-}$ (*slc13a5b*$^{ca205/+}$) zebrafish were generated by using CRISPR/Cas9 mutagenesis. The identified founders carried 11-nucleotide deletion in the fifth exon for *5a* mutations and one-nucleotide substitution followed by 13-nucleotide insertion in the fifth exon for *5b* mutations. The gRNA sequences are CCACACTCACAGGCACTGGACCA for *5a* mutation and GTTTGCTATGCTGCCAGTGTTGG for *5b* mutation. WTs, single homozygous (*5a*$^{-/-}$ and *5b*$^{-/-}$) and double homozygous (*5a*$^{-/-}$;*5b*$^{-/-}$) larvae were used for the experiments (by breeding heterozygous adults). Genotyping to distinguish *5a*$^{-/-}$, *5a*$^{+/-}$ and WTs was carried out by performing high-resolution melt analysis (HRMA) using primers listed in S1 Table. Genotyping to distinguish *5b*$^{-/-}$, *5b*$^{+/-}$ and WTs was carried out by performing HRMA using primers listed in S1 Table, followed by running the non-heterozygous PCR products on agarose gel, based on different sizes of PCR product bands (WTs- 118 bp and *5b*$^{-/-}$- 131 bp). Survival of animals was analyzed by recording their mortality rate every other day, beginning from 0 dpf until 30 dpf and percentage survival was plotted (GraphPad Prism).

### Morpholino injections

The *grin1a* (5′-CCAGCAGAAGCAGACGCATCGT-3′) and *grin1b* (5′-CGAACAGAACCAGGCGCATTTTGCC-3′) MOs were purchased from Genetools (Philomath, OR) and co-injected at the one-cell stage at 0.75 ng concentrations in all experiments described. The standard control MO (5′-CCTCTTACCTCAGTTACAATTTATA-3′) purchased from Genetools was also injected at 0.75 ng concentration. The doses of all MOs were determined as optimal by titration (no toxic effects were observed).

## Gene expression analysis by in situ hybridization and qPCR

Digoxigenin-labeled sense and antisense RNA probes were synthesized for *5a* and *5b* as described previously [66], using the primers listed in S2 Table. The embryos and larvae were fixed at desired stages in 4% paraformaldehyde (Sigma-Aldrich) overnight at 4°C. The whole mount in situ hybridization protocol was performed as described previously [66]. qPCR was performed for *5a*, *5b*, *vglut2a*, *gad1b,* and *fosab* on cDNA obtained from the heads of 5 dpf larvae. *Tg(elavl3:ubci-Cer-sv40)$^{y342}$* animals imported from Zebrafish International Resource Center were outcrossed with *5a$^{+/-}$;5b$^{+/-}$* to raise adults and subsequently *5a$^{-/-}$;5b$^{-/-}$* and WT larvae were obtained for FACS. FACS-sorted neurons were obtained from the pool of heads of 5 dpf larvae *(Tg(elavl3:ubci-Cer-sv40)* and the cDNA obtained was used to perform qPCR for *elavl3*, *gfap*, *5a*, and *5b. rpl13* was used as internal control. The primers used for qPCR are listed in S3 Table.

## Behavioral assays

Zebrafish larvae maintained in 48-well plates were habituated for 20 min, under ambient light. This was followed by behavioral assessment to measure distance moved in 100% light and during acoustic startle, using Zebrabox (Viewpoint Life Sciences). Similarly, larvae were habituated in 20 min of darkness, followed by tracking their movement in 100% darkness. For thigmotaxis protocol, we used 24-well plates and generated a protocol to divide each well into inner well zone and outer well zone. The larvae were habituated for 20 min in ambient light (or treated with 5 mM PTZ prior to this). This was followed by tracking the distance traveled and time spent close to the wall under 6 min of 100% light and 4 min of 100% darkness. Tracking of total distance moved and time taken as a measure of swimming behavior was analyzed using Zebralab V3 software (Viewpoint Life Sciences). Locomotion heatmaps depicting the movement of larvae were also generated by using Zebralab V3 software. All the behavioral assessments were performed on 4 dpf to 6 dpf larvae.

## Drug treatments

Memantine (Cayman Chemical) and MK801 (Sigma-Aldrich) stock solutions were prepared in DMSO. ZnCl$_2$ (Sigma-Aldrich) stock solution was prepared in water. The drugs were assessed for toxicity and the highest concentration which did not induce toxicity, was used in subsequent assays. On the day of experiments, the stock solutions were diluted to a final concentration of 50 μM (for memantine and MK801) and 25 μM (for ZnCl$_2$) in E3 water. The zebrafish larvae were treated with drugs for 2 hours, followed by behavior assessment, Seahorse assay and calcium imaging. The final DMSO concentration was 0.5% and was used as vehicle control in the memantine- and MK801-treatment experiments. Larvae incubated in E3 water alone were used for control experiments for ZnCl$_2$-treated experiments.

## Electrophysiological measurements

Electrophysiological recordings in zebrafish larvae were performed as described previously [41,67]. Briefly, 6 dpf larvae were paralyzed using α-bungarotoxin (1 mg/ml, Tocris) and embedded in 1.2% low melting agarose (LMA). The dorsal side of the larvae was exposed to the agarose gel surface and accessible for electrode placement. Larvae were placed on an upright stage of an Olympus BX51WI upright microscope, visualized using a 5X MPlanFL N Olympus objective, and perfused with embryo media. A glass microelectrode (3–8 MΩ) filled with 2 M NaCl was placed into the optic tectum of zebrafish, and a recording was performed in current-clamp mode, low-pass filtered at 1 kHz, high-pass filtered at 0.1 Hz, using a digital gain of 10 (Multiclamp 700B amplifier, Digidata 1440A digitizer, Axon Instruments) and stored on a PC computer running pClamp software (Axon Instruments). Baseline recording was performed for ≥ five minutes. The threshold for detection of spontaneous events was set at three times noise, as described previously [68]. The frequency and amplitude of spontaneous events were analyzed using the Clampit software 11.0.3 (Molecular Devices, Sunnyvale, CA).

Electrophysiological recordings with NeuroProbe were performed using 5 dpf larvae. Individual larva was transferred to 50 μL E3 water on a clean slide. 2 μL Tricane (0.02%) and 4 μL α-bungarotoxin (1 mg/ml, Tocris) were added to the water

to anesthetize and paralyze the larva for 5 min. The larva was then removed from the anesthetizing solution, embedded in 1.5% low-melting-point agarose with the dorsal side facing upwards, and transferred to the recording stage. The anesthetizing solution was added to the recording bath, in addition to 1,500 μL of E3 water. A 16-channel neural probe (Cambridge NeuroTech, Cambridge, United Kingdom), mounted on a micromanipulator (Luigs and Neumann, Ratingen, Germany), was inserted into the optic tectum of larva. A silver wire was placed in the recording bath as grounding. Physiological signal was amplified using a RHD2132 headstage (Intan Technologies, Los Angeles, United States) and digitized by a USB Acquisition Board (Open Ephys, Altanta, United States) at 30k samples/s. The high-pass was set at 1 Hz and the low-pass set at 7,000 Hz. Data was stored on a computer running the Open Ephys software v 0.5.5.3 [69]. A digital camera (IMX265, Sony Group Corporation, Tokuo, Japan) was used for monitoring and video recording. Each trial lasted ≥ 20 min. Data was subsequently imported into Spike2 v10.21 (Cambridge Electronic Design Limited, Cambridge, United Kingdom) and filtered with a band-pass second order IIR filter, set at 1-1k Hz. Filtered data was used for subsequent analysis and quantification.

## Metabolic measurements

Oxygen consumption rate (OCR) measurements were performed using the XF24 Extracellular Flux Analyzer (Seahorse Biosciences). Individual 6 dpf zebrafish larvae were placed in 24 wells on an islet microplate and an islet plate capture screen was placed over the measurement area to maintain the larvae in place. Seven measurements were taken to establish basal rates, followed by treatment injections and 18 additional cycles [70]. Rates were determined as the mean of two measurements, each lasting 2 min, with 3 min between each measurement [71]. Three independent assays were performed to establish metabolic measurements. A script was generated on R (using dplyr, reshape2, tidyr, purr and tidyverse libraries) to calculate the metabolic measurements for all the parameters. The code for this analysis was uploaded to Zenodo and is available at https://zenodo.org/records/14853182.

## Immunofluorescence

To perform whole mount immunostaining (for HuC/D and pHH3), 3 dpf and 5 dpf larvae were fixed in 4% paraformaldehyde overnight at 4°C, depigmented with a 1% $H_2O_2$/3% KOH solution, followed by antigen retrieval with 150mM Tris HCl (pH 9.0) for 15 min at 68°C, permeabilization (PBS/0.3%Triton-X/1%DMSO/1%BSA/0.1%Tween-20) for 2 hours at room temperature (RT) and blocking (PBS/1%DMSO/2%fetal bovine serum (FBS)/1%BSA/0.1%Tween-20) for 1 hour at RT. Later, the larvae were incubated in primary antibody overnight at 4°C, followed by washing and secondary antibody incubation overnight at 4°C. Finally, the immunostained larvae were washed and counterstained with Hoechst (Thermo Fisher Scientific). pERK/tERK whole mount immunostaining on 5 dpf larvae was performed as described previously [50]. The larvae were mounted in 0.8% LMA for imaging. To perform immunostaining (for HuC/D, Slc13a5, Gfap, vGlut1, Gad67, cleaved-Caspase3, Grin1, NeuN and c-Fos) on 12 μm thick cryosections of 28 hpf embryos, 3 dpf and 5 dpf larvae, antigen retrieval was carried out with citrate buffer (pH 6.0) for 15 min at 95°C. This was followed by permeabilization with 1% Triton-X for 30 min at RT and blocking in 5% normal donkey serum or 5% normal goat serum for 1 hour at RT. Later, the sections were incubated in primary antibody overnight at 4°C, followed by washing and secondary antibody incubation for 3 hours at RT. Finally, the immunostained sections were washed, counterstained with Hoechst and mounted for imaging. The primary antibodies (and their working concentrations) used in this study are listed in S4 Table. Secondary antibodies were used at 1:200 concentration (Life Technologies).

## Live acridine orange staining

The larvae were raised in embryo medium containing 1-phenyly-2-thiourea (PTU) to prevent pigment formation. 5 dpf larvae were incubated for 30 min in embryo medium containing 0.002% A.O. solution (Sigma-Aldrich) at 28°C. The stained fish were washed with embryo medium several times and mounted in 0.8% LMA for live imaging.

 

## Palm-ZP1 and CellMask Orange staining

Palm-ZP1 (dissolved in DMSO) was received as a gift from Stephen J. Lippard lab and used in 5 μM working concentration. CellMask Orange (C10045, Invitrogen) was used as plasma membrane marker in 5 μg/ml working concentration. Fifty hpf embryos were treated with Palm-ZP1 and CellMask Orange for 15 min at 28°C, followed by mounting in 0.8% LMA for live imaging.

## Imaging and quantification

Bright-field images of in situ hybridization-stained embryos and larvae (whole mount and 12 μm thick cryosections) and live animals were obtained using Stereo Discovery.V8 microscope (ZEISS). The live A.O. stained/Palm-ZP1 and CellMask Orange stained/immunostained whole mount samples and cryosections were imaged at 20X magnification using LSM 880/LSM 900 confocal microscopes (ZEISS). Larvae immunostained with pERK/tERK were imaged at 10X magnification in the Airyscan FAST mode of LSM 880. After imaging, the acquired confocal z-stacks were processed, cell counting, and brain width measurements were performed with ZEN software. MB volume measurements were performed using Imaris software. Fluorescence intensity measurements (for Slc13a5, Palm-ZP1, CellMask Orange and Grin1 signal) were performed by calculating CTCF using Fiji software. The ratio of Palm-ZP1:CellMask Orange CTCF was used in data generation.

## Brain activity analysis using pERK/tERK immunostaining

The processing and analysis pipeline is performed using the protocol described previously [49]. Briefly, images were registered to a standard zebrafish reference brain using Computational Morphometry Toolkit (CMTK). The pERK images were normalized with the tERK staining. In MATLAB (MathWorks), the Mann-Whitney U statistic Z score is calculated for each voxel, comparing the mutant and WT groups using MapMAPPING. The significance threshold was set based on a false discovery rate (FDR) where 0.05% of control pixels would be called as significant.

## Calcium imaging and data analysis

*Tg(elavl3:Hsa.H2B-GCaMP6s)$^{if5Tg}$* animals imported from Zebrafish International Resource Center were outcrossed with *5a$^{+/-}$;5b$^{+/-}$* to raise adults and subsequently *5a$^{-/-}$;5b$^{-/-}$* and WT larvae were obtained for calcium imaging. The larvae were raised in E3 medium containing PTU to prevent pigment formation. Briefly, 3 dpf larvae were immobilized using α-bungarotoxin and tricaine and embedded in 0.8% LMA. Larvae were imaged at 10X magnification the Airyscan FAST mode of LSM 880 and single-plane movies were acquired at 9.6 frames per second for ~5 min. Mean grey value measurements of individual neurons in the FB, MB and HB were performed in Fiji and grey values were normalized against a basal mean grey value. A script was generated on R to calculate fluorescence variation measurements (ΔF/F) in each frame of all the neurons, followed by calculating the frequency and amplitude of calcium events (with a threshold value of ΔF/F > 2) for each larvae recorded. The code for this analysis was uploaded to Zenodo and is available at https://zenodo.org/records/14853182. We have used a threshold amplitude value of ΔF/F > 2, considering the pattern that most of the 3 dpf WT larvae stayed within this threshold.

## Bulk RNA sequencing

The heads of 5 dpf larvae were pooled to generate RNA using TRI Reagent (Sigma-Aldrich). Three biological RNA samples were prepared from each genotype, i.e., WTs, *5a$^{-/-}$*, *5b$^{-/-}$* and *5a$^{-/-}$;5b$^{-/-}$*. RNA samples (RIN values >7) were used to generate Stranded poly (A) RNA library and Bulk RNA sequencing (NextSeq2000 P2 100 cycle) was carried out. For RNASeq alignment and differential gene expression analysis, paired end reads were aligned to Danio rerio genome reference GRCz11 release 112 along with annotated transcript GTF file (version 4.3.2) [72] downloaded from Lawson

Lab using STAR (version: STAR_2.5.2b) [73]. Gene counts obtained from STAR were processed to provide an input to DESeq2 (version 1.36.0) for differential gene expression analysis [74]. We excluded those transcripts with zero counts in more than four samples. We used log normalized counts to construct heatmaps for prioritized candidate genes using ComplexHeatmap (v2.12.1) [75,76].

Scripts for differential gene expression analysis and to create heatmaps were uploaded to Zenodo and are available at https://zenodo.org/records/14853182. Sequencing raw and analyzed data are publicly available at NCBI's Gene Expression Omnibus (GEO accession: GSE275235).

## Statistical analysis

GraphPad Prism software was used to perform statistical analysis. Data are represented as mean ± S.E.M. and mean ± S.D. p-values were calculated by unpaired $t$ test. All raw data underlying figures can be found in S1 Data and S2 Data.

## Supporting information

**S1 Fig. Expression analysis of slc13a5 zebrafish paralogs (*5a* and *5b*), transcript and protein level detection in slc13a5 mutants and MB volume measurements.** (A, B) In situ hybridization for *5a* and *5b* expression using sense RNA (control) probes. Negligible expression of *5a* and *5b* indicates that control probes used are reliable. n = 3 for each probe. (C) qPCR analysis for relative *5a* mRNA expression in 5 dpf $5a^{-/-}$, $5b^{-/-}$ and $5a^{-/-};5b^{-/-}$ larvae compared to WTs. *5a* is downregulated in $5a^{-/-}$ and $5a^{-/-};5b^{-/-}$ larvae compared to WTs, indicating active mRNA degradation of $5a^{-/-}$ transcripts. Unchanged *5a* transcript levels in $5b^{-/-}$ larvae points to the absence of genetic compensation by paralog upregulation. WT, $5a^{-/-}$, $5b^{-/-}$ and $5a^{-/-};5b^{-/-}$, n = 3 × 10 larvae pooled. (D) qPCR analysis for relative *5b* mRNA expression in 5 dpf $5a^{-/-}$, $5b^{-/-}$ and $5a^{-/-};5b^{-/-}$ compared to WTs. *5b* is downregulated in $5b^{-/-}$ *and* $5a^{-/-};5b^{-/-}$ larvae compared to WTs, indicating active mRNA degradation of $5b^{-/-}$ transcripts. Unchanged *5b* transcript levels in $5a^{-/-}$ larvae points to the absence of genetic compensation by paralog upregulation. WT, $5a^{-/-}$, $5b^{-/-}$ and $5a^{-/-};5b^{-/-}$, n = 3 × 10 larvae pooled. (E) qPCR analysis for relative mRNA expression of *elavl3* and *gfap* in FACS-sorted neurons obtained from the heads of 5 dpf WTs (*Tg(elavl3:ubci-Cer-sv40))*. *elavl3* is upregulated compared to *gfap*, confirming the purity of the neuronal population. WT, n = 3 × 100 heads pooled. (F and G) qPCR analysis for relative *5a* and *5b* mRNA expression in FACS-sorted neurons obtained from the heads of 5 dpf larvae (*Tg(elavl3:ubci-Cer-sv40))*. *5a* and *5b* transcript levels are significantly reduced in $5a^{-/-};5b^{-/-}$ larvae compared to WTs, further validating that both the paralogs are highly expressed in the neurons compared to astrocytes. WT and $5a^{-/-};5b^{-/-}$, n = 3 × 100 heads pooled. (H-J) 5 dpf WT and $5a^{-/-};5b^{-/-}$ larvae; α-Slc13a5 (green). Quantification of the fluorescence intensity (CTCF) of Slc13a5 in the brain. $5a^{-/-};5b^{-/-}$ larval brains show a significant reduction in the expression of Slc13a5 compared to WTs. WT, n = 6; $5a^{-/-};5b^{-/-}$, n = 6. (K) MB volume quantification at 5 dpf; α-Slc13a5. s*lc13a5* mutants show a significant reduction in the MB volume compared to WTs. WT, n = 7; $5a^{-/-}$, n = 5; $5b^{-/-}$, n = 5; $5a^{-/-};5b^{-/-}$, n = 5. Data are Mean ± S.D., ns: no significant changes observed, *P ≤ 0.05, **P ≤ 0.01, ***P ≤ 0.001- Unpaired $t$ test. The data underlying this figure can be found in S1 Data.
(TIF)

**S2 Fig. Analysis of startle response and thigmotaxis behavior in slc13a5 mutants.** (A) Quantification of distance traveled during complete acoustic startle protocol for 10 min at 5 dpf. slc13a5 mutants move higher distances compared to WTs. WT, n = 21; $5a^{-/-}$, n = 8; $5b^{-/-}$, n = 8; $5a^{-/-};5b^{-/-}$, n = 24. (B) Schematic representation of thigmotaxis protocol. (C) Quantification of distance traveled close to the wall in 10 min. No significant changes were observed in the distance traveled between *slc13a5* mutants and WTs. WT, n = 8; $5a^{-/-}$, n = 6; $5b^{-/-}$, n = 5; $5a^{-/-};5b^{-/-}$, n = 7. (D) Quantification of time spent close to the wall in 10 min. No significant changes were observed in the time spent between *slc13a5* mutants and WTs. WT, n = 8; $5a^{-/-}$, n = 6; $5b^{-/-}$, n = 5; $5a^{-/-};5b^{-/-}$, n = 7. (E) Quantification of distance traveled close to the wall in 10 min post PTZ exposure. *slc13a5* mutants swam significantly higher distance hugging the wall compared to WT. WT, n = 18; $5a^{-/-}$,

n = 8; *5b*⁻/⁻, n = 10; *5a*⁻/⁻*;5b*⁻/⁻, n = 18. (F) Quantification of time spent close to the wall in 10 min post PTZ exposure. *slc13a5* mutants spent significantly more time hugging the wall compared to WT. WT, n = 18; *5a*⁻/⁻, n = 6; *5b*⁻/⁻, n = 10; *5a*⁻/⁻*;5b*⁻/⁻, n = 16. Data are Mean ± S.D., ns: no significant changes observed, *P ≤ 0.05, **P ≤ 0.01, ***P ≤ 0.001- Unpaired *t* test. PTZ, pentylenetetrazol. The data underlying this figure can be found in S1 Data.
(TIF)

**S3 Fig. Neuron population, cell proliferation and apoptosis analysis in slc13a5 mutants.** (A) 3 dpf WTs and *slc13a5* mutants; α-HuC/D (green). (B) 28 hpf WTs and *slc13a5* mutants; α-HuC/D (green), Hoechst (blue). (C) Quantification of HuC/D⁺ cells in the region that gives rise to optic tectum at 28 hpf. HuC/D⁺ cell numbers are unaffected in the *slc13a5* mutants compared to WTs. WT, n = 7; *5a*⁻/⁻, n = 4; *5b*⁻/⁻, n = 4; *5a*⁻/⁻*;5b*⁻/⁻, n = 5. (D-G) 3 dpf and 5 dpf WT and *5a*⁻/⁻*;5b*⁻/⁻ larvae; α-pHH3 (red), Hoechst (blue). Quantification of pHH3⁺ cells in the optic tectum. Cell proliferation is unchanged in the *5a*⁻/⁻*;5b*⁻/⁻ larvae compared to WTs. WT, n = 7; *5a*⁻/⁻*;5b*⁻/⁻, n = 7 at 3 dpf. WT, n = 34; *5a*⁻/⁻*;5b*⁻/⁻, n = 23 at 5 dpf. (H) 5 dpf WTs and *slc13a5* mutants; α-HuC/D (green), CC3 (red). Data are Mean ± S.D., ns: no significant changes observed- Unpaired *t* test. pHH3, Phosphohistone H3. The data underlying this figure can be found in S1 Data.
(TIF)

**S4 Fig. Electrophysiological analysis in 6 dpf.** *5a*⁻/⁻ and *5b*⁻/⁻ larvae. (A-E) Representative extracellular recordings obtained from optic tectum of 6 dpf *5a*⁻/⁻ and *5b*⁻/⁻ larvae, and pie charts of proportion of *5a*⁻/⁻ and *5b*⁻/⁻ larvae showing different patterns of activity. The repetitive inter-ictal like discharges (<1s duration) with above threshold (>0.2mV), high-frequency, large-amplitude spikes seen in *slc13a5* mutants are indicative of increased network hyperexcitability. *slc13a5*⁻/⁻ larvae also displayed below threshold (<0.2mV) brain activity.
(TIF)

**S5 Fig. Electrophysiological analysis in 5 dpf *5a*⁻/⁻*;5b*⁻/⁻ larvae.** (A-D) Representative extracellular recordings obtained using NeuroProbe from optic tectum of 5 dpf *5a*⁻/⁻*;5b*⁻/⁻ larvae, showing epileptic activity, consisting of a series of short bursts, with a total duration of> 10s and repetitive twitches (>5s) at consistent frequency (~1.8 Hz) as another indication of seizures.
(TIF)

**S6 Fig. Bulk RNA sequencing based PCA plot.** PCA plot showing that *5a*⁻/⁻ and *5a*⁻/⁻*;5b*⁻/⁻ are closer to each other compared to WTs and *5b*⁻/⁻ at transcriptomics level. PCA, Principal component analysis.
(TIF)

**S7 Fig. List of upregulated genes in *5a*⁻/⁻ and *5a*⁻/⁻*;5b*⁻/⁻ larvae compared to WTs.** Heatmap showing a list of 400 genes significantly upregulated (padj < 0.05 and log2FC > 1.5) in *5a*⁻/⁻ and *5a*⁻/⁻*;5b*⁻/⁻ larvae compared to WTs. The data underlying this figure can be found in S2 Data.
(TIFF)

**S8 Fig. List of downregulated genes in *5a*⁻/⁻ and *5a*⁻/⁻*;5b*⁻/⁻ larvae compared to WTs.** Heatmap showing a list of 380 genes significantly downregulated (padj < 0.05 and log2FC < −1.5) in *5a*⁻/⁻ and *5a*⁻/⁻*;5b*⁻/⁻ larvae compared to WTs. The data underlying this figure can be found in S2 Data.
(TIFF)

**S9 Fig. List of upregulated genes in *5b*⁻/⁻ larvae compared to WTs.** Heatmap showing a list of 304 genes significantly upregulated (padj < 0.05 and log2FC > 1.5) in *5b*⁻/⁻ larvae compared to WTs. The data underlying this figure can be found in S2 Data.
(TIFF)

**S10 Fig. List of downregulated genes in *5b*<sup>−/−</sup> larvae compared to WTs.** Heatmap showing a list of 444 genes significantly downregulated (padj < 0.05 and log2FC < −1.5) in *5b*<sup>−/−</sup> larvae compared to WTs. The data underlying this figure can be found in S2 Data.
(TIFF)

**S11 Fig. Calcium event analysis in slc13a5 mutants.** (A-F) Representative single neuron calcium traces illustrating that *5a*<sup>−/−</sup>;*5b*<sup>−/−</sup> larvae exhibit above-threshold events in the MB. (G-I) Quantification of the amplitude and frequency of calcium events (ΔF/F > 2) and representative single neuron calcium traces at 3 dpf in the FB. *5a*<sup>−/−</sup>;*5b*<sup>−/−</sup> larvae show significant increase in calcium events compared to WTs. WT, n = 6; *5a*<sup>−/−</sup>;*5b*<sup>−/−</sup>, n = 7. (J-L) Quantification of the amplitude and frequency of calcium events (ΔF/F > 2) and representative single neuron calcium traces at 3 dpf in the HB. *5a*<sup>−/−</sup>;*5b*<sup>−/−</sup> larvae show significant increase in the frequency of calcium events compared to WTs and the amplitude of calcium events remains unchanged. WT, n = 6; *5a*<sup>−/−</sup>;*5b*<sup>−/−</sup>, n = 6. Data are Mean ± S.E.M., ns: no significant changes observed, *P ≤ 0.05, **P ≤ 0.01- Unpaired *t* test. The data underlying this figure can be found in S1 Data.
(TIF)

**S12 Fig. Assessment of the effect of suppressing NMDA receptor signaling and zinc treatment on calcium influx in WTs, and Grin1 expression after grin1a;grin1b MO injection.** (A-B) Quantification of the amplitude and frequency of calcium events (ΔF/F > 2) at 3 dpf in the MB. Memantine treatment and *grin1a;grin1b* MO injections have no significant effect on the calcium events in WTs. Vehicle-treated/control MO-injected WT, n = 5; memantine-treated WT, n = 5; *grin1a;grin1b* MO injected WT, n = 5. (C) Quantification of the fluorescence intensity (CTCF) of Grin1 (α-Grin1). *grin1a;grin1b* MO injections lead to significant reduction in the expression of Grin1 compared to control MO-injected WTs at 3 dpf. control MO-injected WT, n = 5; *grin1a;grin1b* MO-injected WT, n = 6. (D-E) Quantification of the amplitude and frequency of calcium events (ΔF/F > 2) at 3 dpf in the MB. ZnCl$_2$ treatment has no significant effect on the calcium events in WTs. Vehicle-treated WT, n = 4; ZnCl$_2$-treated WT, n = 4. Data are Mean ± S.E.M. and Mean ± S.D., ns: no significant changes observed, *P ≤ 0.05- Unpaired *t* test. The data underlying this figure can be found in S1 Data.
(TIF)

**S13 Fig. Effect of NMDA receptor antagonists and zinc treatment on the bioenergetics parameters of WTs and WT behavior under acoustic startle.** (A) Quantification of basal respiration at 6 dpf. Memantine and MK801 treatments do not affect basal respiration in WTs. Vehicle-treated WT, n = 6; memantine-treated WTs, n = 6; MK801-treated WTs, n = 6 (individual values plotted from five cycles). (B) Quantification of ATP-linked respiration at 6 dpf. Memantine and MK801 treatments do not affect ATP-linked respiration in WTs. Vehicle-treated WT, n = 5; memantine-treated WTs, n = 4; MK801-treated WTs, n = 6 (individual values plotted from three cycles). (C) Quantification of total mitochondrial respiration at 6 dpf. Memantine and MK801 treatments do not affect total mitochondrial respiration in WTs. Vehicle-treated WT, n = 10; memantine-treated WTs, n = 4; MK801-treated WTs, n = 4 (individual values plotted from three cycles). (D) Quantification of non-mitochondrial respiration at 6 dpf. Memantine and MK801 treatments do not affect non-mitochondrial respiration in WTs. Vehicle-treated WT, n = 8; memantine-treated WTs, n = 6; MK801-treated WTs, n = 6 (individual values plotted from three cycles). (E) Quantification of basal respiration at 6 dpf. ZnCl$_2$ treatment does not affect basal respiration in WTs. Vehicle-treated WT, n = 5; ZnCl$_2$-treated WTs, n = 6 (individual values plotted from five cycles). (F) Quantification of ATP-linked respiration at 6 dpf. ZnCl$_2$ treatment does not affect ATP-linked respiration in WTs. Vehicle-treated WT, n = 5; ZnCl$_2$-treated WTs, n = 6 (individual values plotted from three cycles). (G) Quantification of total mitochondrial respiration at 6 dpf. ZnCl$_2$ treatment does not affect total mitochondrial respiration in WTs. Vehicle-treated WT, n = 5; ZnCl$_2$-treated WTs, n = 6 (individual values plotted from three cycles). (H) Quantification of non-mitochondrial respiration at 6 dpf. ZnCl$_2$ treatment does not affect non-mitochondrial respiration in WTs. Vehicle-treated WT, n = 5; ZnCl$_2$-treated WTs, n = 6 (individual values plotted from three cycles). (I) Quantification of total distance traveled in 10 min in acoustic startle before and after treatment with memantine and MK801 at 5 dpf. Memantine

treatment exhibited no effect on the behavior of WTs whereas MK801 treatment led to a significant reduction in the distance traveled by WTs. Vehicle-treated WT, n = 16; memantine-treated WT, n = 19; MK801-treated WT, n = 11. (J) Quantification of total distance traveled in 10 min in acoustic startle before and after treatment with $ZnCl_2$ at 5 dpf. $ZnCl_2$ treatment exhibited no effect on the behavior of WTs. Vehicle-treated WT, n = 9; $ZnCl_2$-treated WT, n = 6. Data are Mean ± S.D., ns: no significant changes observed, ***P ≤ 0.001- Unpaired *t* test. The data underlying this figure can be found in S1 Data. (TIF)

**S1 Movie. Movie (1,000 fps) showing less calcium events in the MB of 3 dpf WT larvae in *Tg(elavl3: Hsa.*H2B-GCaMP6s) background.**
(MP4)

**S2 Movie. Movie (1,000 fps) showing more calcium events in the MB of 3 dpf *5a⁻/⁻;5b⁻/⁻* larvae in *Tg(elavl3*: Hsa.*H2B-GCaMP6s) background.**
(MP4)

**S3 Movie. Movie (1,000 fps) showing less calcium events in the FB of 3 dpf WT larvae in *Tg(elavl3: Hsa.*H2B-GCaMP6s) background.**
(MP4)

**S4 Movie. Movie (1,000 fps) showing more calcium events in the FB of 3 dpf *5a⁻/⁻;5b⁻/⁻* larvae in *Tg(elavl3: Hsa.*H2B-GCaMP6s) background.**
(MP4)

**S5 Movie. Movie (1,000 fps) showing less calcium events in the HB of 3 dpf WT larvae in *Tg(elavl3:Hsa. H2B-GCaMP6s)* background.**
(MP4)

**S6 Movie. Movie (1,000 fps) showing more calcium events in the HB of 3 dpf *5a⁻/⁻;5b⁻/⁻* larvae in *Tg(elavl3: Hsa.*H2B-GCaMP6s) background.**
(MP4)

**S7 Movie. Movie (1,000 fps) showing more calcium events in the MB of 3 dpf vehicle-treated/control MO-injected *5a⁻/⁻;5b⁻/⁻* larvae in *Tg(elavl3:Hsa.*H2B-GCaMP6s) background.**
(MP4)

**S8 Movie. Movie (1,000 fps) showing less calcium events in the MB of 3 dpf memantine-treated *5a⁻/⁻;5b⁻/⁻* larvae in *Tg(elavl3:Hsa.*H2B-GCaMP6s) background.**
(MP4)

**S9 Movie. Movie (1,000 fps) showing less calcium events in the MB of 3 dpf *grin1a;grin1b* MO-injected *5a⁻/⁻;5b⁻/⁻* larvae in *Tg(elavl3:Hsa.*H2B-GCaMP6s) background.**
(MP4)

**S10 Movie. Movie (1,000 fps) showing more calcium events in the MB of 3 dpf vehicle-treated *5a⁻/⁻;5b⁻/⁻* larvae in *Tg(elavl3:Hsa.*H2B-GCaMP6s) background.**
(MP4)

**S11 Movie. Movie (1,000 fps) showing less calcium events in the MB of 3 dpf $ZnCl_2$-treated *5a⁻/⁻;5b⁻/⁻* larvae in *Tg(elavl3:Hsa.*H2B-GCaMP6s) background.**
(MP4)

**S1 Data.  Raw numerical data.**
(XLSX)

**S2 Data.  Raw data for Bulk RNA sequencing.**
(XLSX)

**S1 Table.  Primer list used for genotyping.**
(XLSX)

**S2 Table.  Primer list used to generate in situ hybridization probes.**
(XLSX)

**S3 Table.  Primer list used in qPCR.**
(XLSX)

**S4 Table.  Primary antibodies used in immunofluorescence.**
(XLSX)

## Acknowledgments

We would like to thank Boogyung Seo for help with behavioral analyses. We also thank Rachel Lacroix for training on pERK/tERK data processing. We further thank Sisu Han, Faizan Malik, Eric Samarut and Andrew Boyce for suggestions and training on calcium imaging processing. We also thank Pia Svendsen (ACHRI) for training on confocal microscopes. We thank HBI AMP facility for training on Imaris software. We thank Stephen J. Lippard and Jacob Goldberg for sharing Palm-ZP1 reagent. We also thank DNA sequencing facility and flow cytometry facility (University of Calgary) for providing bulk RNA sequencing and FACS-sorting services, respectively. We also thank the Peng Huang lab for sharing their microinjection facility. We further thank Arthur Omorogiuwa and Katrinka Kocha for assistance with zebrafish husbandry. We thank Natasha Klenin for assistance with reagent preparation. We also thank Anneke Korver and Kyle Morin for help with adult zebrafish genotyping for colony maintenance. We thank Alicia Vandenbrink for administrative support. Finally, we thank TESS Research Foundation and their families for their continued interest in our zebrafish program.

## Author contributions

**Conceptualization:** Deepika Dogra, Deborah M Kurrasch.

**Data curation:** Deepika Dogra, Van Anh Phan, Sinan Zhang, Cezar Gavrilovici, Nadia DiMarzo, Ankita Narang, Kingsley Ibhazehiebo, Deborah M Kurrasch.

**Formal analysis:** Deepika Dogra, Deborah M Kurrasch.

**Funding acquisition:** Deepika Dogra, Deborah M Kurrasch.

**Investigation:** Deepika Dogra, Van Anh Phan, Sinan Zhang, Cezar Gavrilovici, Nadia DiMarzo, Ankita Narang, Kingsley Ibhazehiebo, Deborah M Kurrasch.

**Methodology:** Deepika Dogra.

**Supervision:** Deborah M Kurrasch.

**Validation:** Deepika Dogra.

**Writing – original draft:** Deepika Dogra.

**Writing – review & editing:** Deepika Dogra, Deborah M Kurrasch.

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
