## [Editor Report · Decision Letter 0]

12 Jan 2024

Dear Dr Kurrasch, 

Thank you for submitting your manuscript entitled "Epileptic phenotypes in slc13a5 loss-of-function zebrafish are rescued by blocking NMDA receptor signaling" for consideration as a Research Article by PLOS Biology.

Your manuscript has now been evaluated by the PLOS Biology editorial staff as well as by an academic editor with relevant expertise and I am writing to let you know that we would like to send your submission out for external peer review.

Once your full submission is complete, your paper will undergo a series of checks in preparation for peer review. After your manuscript has passed the checks it will be sent out for review. To provide the metadata for your submission, please Login to Editorial Manager (https://www.editorialmanager.com/pbiology) within two working days, i.e. by Jan 16 2024 11:59PM.

Kind regards,

Lucas

Lucas Smith, Ph.D.

Senior Editor

PLOS Biology

lsmith@plos.org

---

## [Decision Letter · Decision Letter 1]

8 Mar 2024

Dear Dr Kurrasch,

Thank you for your patience while your manuscript "Epileptic phenotypes in slc13a5 loss-of-function zebrafish are rescued by blocking NMDA receptor signaling" was peer-reviewed at PLOS Biology. Your manuscript has been evaluated by the PLOS Biology editors, an Academic Editor with relevant expertise, and by several independent reviewers.

The reviewer reports can be found at the end of this email. Although the reviewers find the work potentially interesting, and while they are somewhat mixed in the amount of work requested before publication, overall we think concerns raised by the reviewers are important and that they would need to be thoroughly addressed before we can consider your manuscript for publication. Based on their specific comments and following discussion with the Academic Editor, it is clear that a substantial amount of work would be required to meet the criteria for publication in PLOS Biology. However, given our and the reviewer interest in your study, we would be open to inviting a comprehensive revision of the study that thoroughly addresses all the reviewers' comments. Given the extent of revision that would be needed, we cannot make a decision about publication until we have seen the revised manuscript and your response to the reviewers' comments. Your revised manuscript would need to be seen by the reviewers again, but please note that we would not engage them unless their main concerns have been addressed. 

Having discussed the reviews with the Academic Editor we would like to emphasize the need to conduct further work to characterize this model, including its epilepsy phenotypes as the reviewers suggest. We also agree with reviewer 4 that additional work is needed to interrogate and better support the zinc chelator hypothesis. We think these revisions are important for moving the field forward and to enhance the utility of this model. 

We appreciate that these requests represent a great deal of extra work, and we are willing to relax our standard revision time to allow you 6 months to revise your study. Please email us (plosbiology@plos.org) if you have any questions or concerns, or envision needing a (short) extension.

**IMPORTANT - SUBMITTING YOUR REVISION**

*Resubmission Checklist*

*Published Peer Review*

*PLOS Data Policy*

*Blot and Gel Data Policy*

Sincerely,

Luke

Lucas Smith, Ph.D.

Senior Editor

PLOS Biology

lsmith@plos.org

REVIEWS:

Reviewer #1: This manuscript from Dogra et al. reports the phenotypic characterization of a zebrafish model of slc13a5 inactivation. 

The study is well constructed, and it targets an important area of biomedical research. Murine models of Slc13a5 inactivation do not provide significant translational information. As such, the introduction of alternative models is welcome. 

The Authors provide a very nice description of the phenotype with behavior, cellular and metabolic assessment. They also show convincing data to link NMDA imbalance with the epilepsy-related phenotype. Final the study highlights a divergence between the epilepsy- and developmental-linked events.

Reviewer #2, Emre Yaksi (note, reviewer 2 has signed this review): In this study from the Kurrasch laboratory, the authors characterize zebrafish mutants for slc13a5, a citrate transporter. The authors demonstrate that the animals exhibit hyperactive behaviors, and physiological data supports increased neural activity and decreased metabolic rate. The authors also reveal increased PERK in the forebrain and present evidence for elevated calcium activity, although the location and nature of calcium signals are less clear. In the final section, the authors employ morpholinos (MO) against glun1 to illustrate that targeting NMDA receptors can rescue some of the neural excitability, behavioral, and metabolic phenotypes.

I believe that this study provides a comprehensive characterization of the mutant line, showcasing hyperactive behaviors, increased neural activity, and reduced metabolism. However, based on the presented data, I am unsure whether we are observing a seizure/epilepsy model or a hyperactivity phenotype. While animals with such high levels of activity might eventually experience seizures, the data at this developmental stage does not resemble a typical seizure phenotype, as expected from other zebrafish and epilepsy models. In my view, this study is interesting for understanding the role of citrate transporters in brain function. However, I think the authors need to better characterize their animals. If the authors claim that these animals truly experience seizures, they should demonstrate this seizure phenotype with more robust physiological and calcium experiments.

I am optimistic about the value of this study, and I truly hope some of my comments will assist the authors in improving their research.

MAJOR Comments:

1. One major concern is the authors' claim that these mutants are an epilepsy/seizure model. Clear evidence of seizures in these animals is missing. Performing calcium imaging and analyzing the data to meet standards of the field should clarify this. The examples in Figure 3D appear to be interictal activity, but have these mutant animals ever shown large ictal bursts followed by long depression periods? I could not find any evidence of such seizure-like ictal activity in the electrical or calcium recordings from mutant animals. Therefore, I am convinced that these animals are hyperactive, both behavioral, neural and metabolic recordings points towards this, but I did not find what evidence is presented for observed seizures in these mutant zebrafish. Please see these papers, how such large ictal bursts may look during ictal bursts of calcium and electrical recordings: https://doi.org/10.1038/s41467-019-11739-z , https://doi.org/10.1002/glia.24106 , https://doi.org/10.1111/epi.17380

2. I would like to see the electrophysiological and calcium data from individual animals presented more effectively. As of now, we are asked to rely on the authors' classification and a few example traces, but it is crucial to present data from individual animals, at least as a supplemental figure. This would allow a closer examination of the potential causes of such large-scale variability across animals. I would also appreciate an explanation why many mutants do not exhibit physiological phenotypes, while mutant behavior seems to show a big differences from controls.

3. The authors make an argument towards unchanged neurogenesis but increased cell death. I am fine but in that case the authors needs to make more a solid quantification of neurogenesis. For this the authors can use Phosphorylated Histone H3 staining. effective anti-pH3 antibody staining can be found here: https://doi.org/10.1101/2024.02.01.578354 or here https://doi.org/10.1126/sciadv.aaz3173 . PH3 staining will label diving cells, and I would recommend to do this staining at few different stages between 2-5 days.

4. The increase in amplitude and frequency of calcium events in mutants is interesting, but it is not clear how these events are detected, based on what criteria, and how robust and comparable these are across animals and conditions. More data presentation of such detected events over calcium traces would be helpful. Defining event thresholds in unbiased ways and performing complementary analyses would provide more confidence in the robustness of these findings.

5. The brain regions analyzed for calcium signals (presumably the optic tectum) need clarification. Are these signals, or at least conclusions of the results, similar in other brain regions such as the forebrain and brainstem? The authors showed a significant increase in pERK signaling in the forebrain but not in the midbrain and hindbrain of mutants. It is unclear if these regions with the largest increase in neural activity also show the largest PERK signals and the largest amount of neural death. This link is not clearly shown or explained.

6. It is not clear to me how hyperactive mutant animals with more robust swim behaviors exhibit lower levels of metabolic and basal respiration signals. The mechanisms explaining this contradiction need clarification.

7. The MO experiments in Figure 4E-H suggest a role of these proteins in neural excitability. However, presume the role of Glun1 in neural excitbity is well known. So it is not clear what we are learning from these experiments and how it relates to the remaining part of this study ?

8. What experiments demonstrate a role for NMDA receptors in slc13a5 mutant dysregulation of Ca+2 signalling ? I guess for making this statement one needs to do transcriptome or proteome analysis, isn't it ?

MINOR Comments:

1. In general, slc13a5 mutants move a lot more even without acoustic stimulation. Therefore, the phenotype looks more like increased swim behavior rather than increased acoustic startle. To show increased acoustic startle, the authors should demonstrate and quantify transient increases in swim behaviors upon acoustic stimulation compared to the baseline period preceding the acoustic stimulation.

2. For circadian behaviors, please display the time course of day/night swim cycles for the recorded animals rather than just the average swim distance during the night. Please refer to Jason Rihel's paper for an example of how to display such sleep data.

3. Please note the concentration of PTZ challenge in figure S2.

4. Do the results of Figure 3A/B also correlate with the general ratio/distribution of glutamatergic and GABAergic neurons in the brain?

5. The authors wrote, "We speculate that a range of electrical activities, i.e., above and below threshold spikes, observed in our mutants could be because of the stochastic nature of seizures." It is not clear what is meant by this statement and how it explains the lack of evidence for large ictal bursts followed by postictal depression.

6. The sentence "Pharmacologically, we exposed WT and mutant zebrafish larvae to meantime, an established NMDA antagonist used in other zebrafish studies" seems incomplete. Additionally, is there a figure for the drug experiment? Please name the drug in the main text.

Reviewer #3, Quynh Anh Nguyen (note, reviewer 3 has signed this review): Dogra et al. characterize their newly created zebrafish model of SLC13A5 deficiency and show that it displays many behavioral features seen in humans with mutations of this gene. They find that their slc13a5 mutants show increased neuronal excitation, with increased glutamatergic and decreased GABAergic gene expression, manifesting in the occurrence of inter-ictal like discharges seen in a subset of their mutant animals. The authors go on to show that increased NMDA receptor function underlies the abnormal calcium activity and behavioral phenotypes observed, demonstrating rescue of these phenotypes through either pharmacological or genetic blockade of NMDA receptors. Altogether, this study is novel and innovative, providing a valuable zebrafish model to use in studying a disease mechanism for which prior mouse models had significant limitations, and identifying a possible mechanism for the etiology of epilepsy associated with SLC13A5 deficiency in humans. The following major comment and minor comments will help provide further clarification and enhance the study for publication in PLOS Biology.

Major comment:

1) The main limitation of this model is the limited epilepsy phenotype displayed, with only a minority of mutants showing inter-ictal like discharges and no seizures observed in any of the mutants. In addition, it is unclear to what extent the calcium events imaged correlate to this abnormal electrophysiological activity (is it possible to do both calcium imaging and extracellular field recordings simultaneously?). Furthermore, the title of the study states that the epileptic phenotypes are rescued by NMDA blockade, but the authors only show rescue specifically for the calcium events, bioenergetics, and startle response and do not show whether there are less inter-ictal like discharges. Given the limitations with studying the spontaneous aspects of epilepsy in this model, notably seizures, it would be worth assaying whether the mutants show increased seizure susceptibility to PTZ exposure (do they display a more severe seizure phenotype or are more susceptible to seizures?). If they do, then this could be used to test whether NMDA blockade is able to decrease this seizure susceptibility and provide more support for their conclusion that this mechanism is responsible for the epilepsy-related phenotypes in their model.

Minor comments:

2) The images in Fig. S1A and B were taken at 3dpf and additional images taken at 5dpf should be provided to help with comparisons.

3) Quantification of the cell type characterization in Fig. 1C and D should be provided.

4) Fig. 2, panels G and H: the increase in number of HuC/D+ cells between 3 and 5dpf across all genotypes suggests that there is still neurogenesis occurring even in the mutants during this time period. Are there any markers they could stain for to check that the rate of neurogenesis is not altered between WT and mutants and could contribute to the decrease in neurons seen in the mutants?

5) Why was calcium imaging done at 3dpf instead of at 5dpf, which would be more consistent with the timeline of their other assays?

6) More n's are needed for Fig. S6A and B.

7) Does knockout of only 5a expression affect 5b expression and vice versa?

Reviewer #4: Dogra et al. use zebrafish to model a form of infantile epileptic encephalopathy caused by mutations in the sodium-dependent citrate transporter slc13a5, with the goal of discovering disease etiology. The authors make single and double CRISPR knock-outs of paralogs slc13a5a and slc13a5b which survive the early larval period but die by 30 days of age. They describe a range of brain phenotypes: reduced brain size, reduced neuron number, increased neuronal cell death, increased behavioral excitability and motility, increased synthesis of excitatory and decreased synthesis of inhibitory neurotransmitters, increased overall neuronal activity, and generally reduced metabolic flux. 

Based on their findings, the authors propose a model whereby increased extracellular citrate levels in the absence of the slc13a5 transporter chelates zinc which is normally required to regulate NMDA signaling. Consistent with this, knocking down NMDARs pharmacologically or with antisense Morpholinos rescues the slc13a5 double mutant phenotype. 

The paper introduces a new animal model that recapitulates aspects of the epileptic phenotype of humans with mutations in slc13a5. However, the phenotypic characterization is superficial and the connection between the various aspects of the mutant phenotype needs to be strengthened. The robust suppression of the phenotype by NMDA block is intriguing but it is insufficient support for the zinc chelator model, since suppressing a citrate-induced increase in neuronal activity and behavior by blocking NMDA could be quite indirect. While this new animal model may be developed into a tool for further research into epileptic phenotypes, as it stands the paper fails to sufficiently elucidate the etiology of the disorder for publication in PLoS Biology.

Specific criticisms: 

1) the zinc chelator hypothesis is not sufficiently well supported by the finding that blocking NMDA rescues slc13a5 mutant phenotypes. A more direct test of the role of zinc as a key component of the pathway is required. 

2) The stated rationale for knocking both slc13a5a and a5b down is that they may function redundantly or partly redundantly. This predicts more severe phenotypes in the double mutant than either single mutant. Regarding lethality this is observed: double mutants die sooner than either single mutant. However with respect to brain size, neuron numbers, cell death and vglut2a expression the double mutant is the same as either single mutant. Inexplicably, with respect to behavior the single slc13a5b mutant is more excitable than the double mutant. Most strangely of all, with respect to gad1b expression and electrophysiological activity, double mutants are much less severely affected than either single mutant. The authors do not note these unexpected genetic relationships and haven't done the statistics on them, but they are very obvious and need to be explained. 

3) The characterization of slc13a5a/b expression is superficial. Slc13a5 expression is stated to be enriched in the midbrain (Fig. 1A, S1A,B) but in fact appears to be throughout the brain; a sense control probe should be included in Fig. 1A (mysteriously, two scRNA-Seq atlases show little (a5a) or no (a5b) expression of either paralog in the brain). Later, using a polyclonal rabbit antibody raised against human Slc13a5, they show expression is shown to be quite specific in a subset of neurons and glial cells (Fig. 1C, D, Fig. 4E) but what and where these neurons are is not determined. First, the specificity of the antibody for zebrafish slc13a5 proteins should be validated by staining of single and double mutants. Then the specific expressing neurons should be identified and shown to be relevant to the other phenotypes. (Additional note: the Grin1 expression in the neurons in Fig. 4E is not detectable; separate channels should be shown). 

4) The description of mutant phenotypes is disjointed and superficial. Which cells in the midbrain are dying and missing? Are they the same neurons (or astrocytes?) that express slc13a5? Which neurons show increased GCaMP signals? Are they the same as the slc13a5-expressing neurons? What aspect of midbrain size is reduced? The measurement is of an (unstated) linear dimension whereas a volume or at least an area would more fully capture the size of the regions. The metabolic analysis is also very superficial, being of whole fish, while the metabolic abnormalities of the epileptic brain are specifically in the brain. Some correlate of metabolic defects in the brain, and specifically in and around the slc13a5-expressing neurons should be included. Finally, the model is that increased excitability is the direct consequence of excess extracellular citrate causing increased NMDA signaling. Yet the authors also describe a strong increase in glutamate synthesis (vgut2 expression). What is the connection between the two excitatory phenotypes?

---

## [Decision Letter · Decision Letter 2]

4 Dec 2024

Dear Dr Kurrasch,

Thank you for your patience while we considered your revised manuscript "Modulation of NMDA receptor signaling and zinc chelation prevent seizure-like events in slc13a5 loss-of-function zebrafish" for publication as a Research Article at PLOS Biology. Your revised study has been evaluated by the PLOS Biology editors, the Academic Editor and by original reviewers 2-4. 

The reviews are appended below. As you will see the reviewers agree that the study has been strengthened in the last revision. Reviewers 2 and 3 suggest that we accept the study, after a few minor changes. However, Reviewer 4 has a number of important lingering concerns about some of the data which we think will need to be addressed before publication. For the most part, we think that Reviewer 4's comments can be addressed with textual changes , however we think that some additional data, such as more compelling palm-ZP1 quantification and assessment of Slc13a5 by western blot, may be needed to address reviewer 4's concerns. 

While we think the scope of the remaining revisions needed is relatively straightforward, given that some new data is needed and given the upcoming holidays, we have provided a 3 month deadline for the next revision. Please email us (plosbiology@plos.org) if you have any questions or concerns, or if you end up needing an extension. 

Given the extent of revision needed, we cannot make a decision about publication until we have seen the revised manuscript and your response to the reviewers' comments. Your revised manuscript may be sent out for further evaluation by a subset of the reviewers.

**IMPORTANT - SUBMITTING YOUR REVISION**

*Re-submission Checklist*

*Published Peer Review*

*PLOS Data Policy*

*Blot and Gel Data Policy*

Sincerely,

Lucas

Lucas Smith, Ph.D.

Senior Editor

PLOS Biology

lsmith@plos.org

REVIEWS:

Reviewer #2, Emre Yaksi (note, reviewer 2 has signed this review): I thank the authors for all additional work and clarifications. The revised manuscript has addressed my comments. My only remaining comment is that the text in some of the updated figures appears very small. I would highly recommend the authors for an update in some of their figure organization and text size. 

Reviewer #3, Quynh Anh Nguyen (note, reviewer 3 has signed this review): Dogra et al. provide a comprehensive analysis of their zebrafish model of SLC13A5 deficiency and show the involvement of NMDA signaling and zinc chelation in the emergence of epilepsy-like phenotypes. This new zebrafish model may be useful in future drug discovery studies to identify more effective treatments for the associated disease in humans. 

The authors have addressed my concerns, especially providing more convincing data supporting the epilepsy-like seizure phenotype in their model. Although they still find that a majority of mutant animals do not display epileptic-like activity even with an additional more sensitive electrophysiological measure (Fig 3M), this model may still be useful for future drug screening studies. A remaining question is whether there is correlation between animals that display electrophysiological abnormal activity and the behaviors assayed, which will help provide a simple way to screen epileptic-like from non-epileptic like individuals in the mutants.

One minor suggestion: please add a label to Fig 3M to distinguish it from Fig 3I, maybe a label stating NeuroProbe.

Reviewer #4: The authors have substantiated the argument that the citrate transporter slc13a5 neuronal phenotype is caused by reduced extracellular zinc in two ways. 1) By measuring extracellular zinc (Palm-ZP1 staining) and 2) by showing that supplementing with ZnCl2 rescues many of the phenotypes. The ZnCl2 supplementation results are compelling and significantly strengthen the manuscript (Fig. 7K-P). However, the Palm-ZP1 quantitation (simply comparing levels of green fluorescence in WT vs mutant) is not done appropriately for several reasons: 1) the dye doesn't appear to penetrate far into the embryo, so the staining appears to be of superficial epithelial cells, not neurons. 2) the quantitation is simply of the Palm-ZP1 channel. The authors should compare the green:red ratio (Palm-ZP1:membrane stain) in WT and mutants. 

There remain several other concerns about the quality of the work that should also be addressed, though largely without further experiments: 

1) The RNA in situs in Fig. 1A-F in no way show "restriction" of slc13a5 to the midbrain. Expression is widespread in the brain and the head periphery, with slightly stronger expression in the tectum (as well as other structures). The authors should revise this statement. Also, the fact that slc13a5 gene expression is detected in FACs-purified Elavl3-expressing cells is evidence that slc13a5 genes are expressed in neurons but it is not "evidence for neuron-specific expression". This statement should be revised. 

1) The immunostaining of Slc13a5 is only reduced by 50% in double mutants (Fig. S1J). What is the antibody raised against? The methods say it was raised in rabbit but against which species' Slc13a5? N- or C-terminal peptides? If it was raised against a C-terminal peptide then it means that the remaining 50% of the staining detected in double mutants is non-specific. A western blot would be more quantitative and would determine if any full-length protein persists in mutants.

2) The immunostaining in Fig. 1G-J is undecipherable. Is the Slc13a5 the green speckles? Given that 50% of the staining may be non-specific (my point above), these images do not add any valuable information. Also, there is no mention of the co-staining with vGlut1 and Gad67 (Fig. 1I,J) and since these images are uninterpretable, these panels should be removed. Note: the legend says that there is Hoechst staining, but it is not shown in the image. 

3) The authors have added RNA-Seq comparing WT, single and double mutants with the conclusion that slc13a5a contributes more to the double mutant phenotype than slc13a5b. However apart from the PCA plot, which is an abstraction, it is impossible to assess this assertion because the heat maps in Fig. 5 and Fig. S7 don't include all four genotypes. There should be counts for all the genes shown in all of the genotypes, so there is no reason to split out the slc13a5b data from the slc13a5a and double mutant data. All the affected genes should be shown for all four genotypes in the same heat map.

4) Fig. 2F,G shows significant reduction in the number of neurons in the tectum, but Fig. 6 H,I, using a different marker of mature neurons, NeuN, shows a significant increase in the number of neurons (and a corresponding increase in grin1 expression). How are these contradictory results reconciled? 

5) The mutant generation and genotyping section of the methods says "WTs, single homozygous (5a-/- and 5b-/-) and double homozygous (5a-/-;5b-/-) larvae were used for the experiments (by breeding heterozygous adults)". This means that for each double mutant "n" in the manuscript, of which there are a lot, ~16 animals would have been assayed and subsequently genotyped. Is this really the case? If so, there should be vastly more WT and single mutant n's than double mutant n's but this doesn't appear to be the case. If, in fact, homozygous mutants were bred together for experiments, what was the source of WT controls? How were the experimenters blinded to the genotypes of the animals they were assaying? 

6) The methods section for electrophysiology says that animals were anesthetized with Tricaine and bungarotoxin "to anesthetize and paralyze" the larvae and that the anesthetic was added to the recording bath. How then do the animals generate the twitches quantified in Fig. 3? It isn't valid to quantify body movement under paralyzing conditions, since any differences between WT and mutants may reflect a resistance to anaesthetic/paralytic in mutants rather than an intrinsic increased excitability. 

7) Statistics: Two different symbols are used to indicate P<0.01: ** and ##. What is the difference between them? Please clarify.

---

## [Editor Report · Decision Letter 3]

11 Feb 2025

Dear Dr Kurrasch,

Thank you for your patience while we considered your revised manuscript "Modulation of NMDA receptor signaling and zinc chelation prevent seizure-like events in slc13a5 loss-of-function zebrafish" for publication as a Research Article at PLOS Biology. This revised version of your manuscript has been evaluated by the PLOS Biology editors and the Academic Editor who is satisfied by the changes made in response to the previous round of review.

Based on our Academic Editor's assessment of your revision we are likely to accept this manuscript for publication. However, before we can accept your study we need you to address a number of data and other policy-related requests, in a last, short revision. These are detailed below. 

**IMPORTANT: Editorial requests

1) TITLE: we would like to propose a tweak to the title. If you agree, we suggest the title be changed to: 

'Modulation of NMDA receptor signaling and zinc chelation prevent seizure-like events in a zebrafish model of slc13a5 epilepsy'

or even 

'Modulation of NMDA receptor signaling and zinc chelation prevent seizure-like events in a zebrafish model of epilepsy' 

We think that these changes will make the title more broadly accessible

2) DATA: You may be aware of the PLOS Data Policy, which requires that all data be made available without restriction: http://journals.plos.org/plosbiology/s/data-availability. For more information, please also see this editorial: http://dx.doi.org/10.1371/journal.pbio.1001797

a. Supplementary files (e.g., excel). Please ensure that all data files are uploaded as 'Supporting Information' and are invariably referred to (in the manuscript, figure legends, and the Description field when uploading your files) using the following format verbatim: S1 Data, S2 Data, etc. Multiple panels of a single or even several figures can be included as multiple sheets in one excel file that is saved using exactly the following convention: S1_Data.xlsx (using an underscore).

b. Deposition in a publicly available repository. Please also provide the accession code or a reviewer link so that we may view your data before publication. 

>>Regardless of the method selected, please ensure that you provide the individual numerical values that underlie the summary data displayed in the following figure panels as they are essential for readers to assess your analysis and to reproduce it:

Fig 1K,U-X; Fig 2B,D,F-G,I-J; Fig 3A-C,E; Fig 4A-H; FIg 5A-B; Fig 6A-C,J-K;QR; Fig 7A-P; Fig S1C-G,J-K; Fig S2 A,C-F; Fig S3C,F-G; FIg S11 G-H,J-K; Fig S12A-E; Fig S13A-J;

>>Please also ensure that figure legends in your manuscript include information on where the underlying data can be found, and ensure your supplemental data file/s has a legend.

>>Please ensure that your Data Statement in the submission system accurately describes where your data can be found.

3) DATA AVAILABILITY STATEMENT: Thank you for uploading your raw RNA-seq data to GEO. Can you please reference this deposition in your data availability statement in the relevant section of the Editorial Manager system? Please note that this data will need to be made publicly available before publication. 

4) CODE: Per journal policy, if you have generated any custom code during the course of this investigation, please make it available without restrictions. Please ensure that the code is sufficiently well documented and reusable, and that your Data Statement in the Editorial Manager submission system accurately describes where your code can be found. 

We expect to receive your revised manuscript within two weeks. 

*Published Peer Review History*

*Press*

Sincerely,

Luke

Lucas Smith, Ph.D.

Senior Editor

lsmith@plos.org

PLOS Biology

---

## [Editor Report · Decision Letter 4]

26 Feb 2025

Dear Dr Kurrasch,

Thank you for the submission of your revised Research Article "Modulation of NMDA receptor signaling and zinc chelation prevent seizure-like events in a zebrafish model of SLC13A5 epilepsy" for publication in PLOS Biology, and thank you for addressing our editorial requests in this revision. On behalf of my colleagues and the Academic Editor, Eunjoon Kim, I am pleased to say that we can in principle accept your manuscript for publication, provided you address any remaining formatting and reporting issues. These will be detailed in an email you should receive within 2-3 business days from our colleagues in the journal operations team; no action is required from you until then. Please note that we will not be able to formally accept your manuscript and schedule it for publication until you have completed any requested changes.

PRESS

Sincerely, 

Luke

Lucas Smith, Ph.D.

Senior Editor

PLOS Biology

lsmith@plos.org